DOI: 10.1038/s41467-017-01658-2　　**OPEN**

# Regulation of gene expression and RNA editing in *Drosophila* adapting to divergent microclimates

Arielle L. Yablonovitch[1,2], Jeremy Fu[1], Kexin Li[3,4], Simpla Mahato[5], Lin Kang[6], Eugenia Rashkovetsky[3], Abraham B. Korol[3], Hua Tang[1], Pawel Michalak[6,7,8], Andrew C. Zelhof[5], Eviatar Nevo[3] & Jin Billy Li [1,2]

Determining the mechanisms by which a species adapts to its environment is a key endeavor in the study of evolution. In particular, relatively little is known about how transcriptional processes are fine-tuned to adjust to different environmental conditions. Here we study *Drosophila melanogaster* from 'Evolution Canyon' in Israel, which consists of two opposing slopes with divergent microclimates. We identify several hundred differentially expressed genes and dozens of differentially edited sites between flies from each slope, correlate these changes with genetic differences, and use CRISPR mutagenesis to validate that an intronic SNP in *prominin* regulates its editing levels. We also demonstrate that while temperature affects editing levels at more sites than genetic differences, genetically regulated sites tend to be less affected by temperature. This work shows the extent to which gene expression and RNA editing differ between flies from different microclimates, and provides insights into the regulation responsible for these differences.

[1] Department of Genetics, Stanford University School of Medicine, Stanford, CA 94305, USA. [2] Biophysics Program, Stanford University, Stanford, CA 94305, USA. [3] Institute of Evolution, University of Haifa, Haifa 3498838, Israel. [4] Institute of Apicultural Research, Chinese Academy of Agricultural Sciences, 100093 Beijing, China. [5] Department of Biology, Indiana University, Bloomington, IN 47405, USA. [6] Edward Via College of Osteopathic Medicine, Blacksburg, VA 24060, USA. [7] Biocomplexity Institute, Virginia Tech, Blacksburg, VA 24061, USA. [8] Center for One Health Research, Virginia-Maryland College of Veterinary Medicine, Blacksburg, VA 24060, USA. Correspondence and requests for materials should be addressed to E.N. (email: nevo@research.haifa.ac.il) or to J.B.L. (email: jin.billy.li@stanford.edu)

Gene expression and RNA editing are two dynamic transcriptional processes that can potentially be altered through the process of natural selection, but the prevalence and mechanisms of this phenomenon have only begun to be explored relatively recently. Evidence suggests that changes in gene expression can be adaptive, in species ranging from humans to fruit flies, and fish to yeast[1–5]. In many of these cases, a mutation in the regulatory region of the gene of interest has been identified as the cause of the gene expression change. It is clear that gene expression is a trait that is selected for in different species and environments, and may even play a stronger role in adaptation than changes in the amino acid sequences of proteins[1].

While there have been several promising studies highlighting gene expression adaptation, much less is known about how RNA editing levels are altered in response to changing environments. The most common form of RNA editing is Adenosine-to-Inosine, or A-to-I editing, catalyzed by adenosine deaminase acting on RNA (ADAR). Inosine gets recognized as guanosine during translation, so A-to-I RNA editing has the potential to alter amino acid sequences of proteins and affect other transcriptional processes[6–8]. It is especially important for proper neuronal function; for instance, mice lacking ADAR2 die of seizures early in life[9], and flies lacking ADAR display a host of neurological phenotypes, including age-dependent neurodegeneration, problems with locomotion, as well as courtship and circadian rhythm defects[10,11]. In light of these findings, it is interesting that one of the major studies that showed an adaptive role for RNA editing involved a non-synonymous editing site in a potassium channel[12]. In that study, octopuses that lived in warmer temperatures had lower editing levels at this site, while those that lived in cooler water temperatures had higher editing levels. The levels of this site were also shown to affect the gating kinetics of the potassium channel and help explain how the channel functions properly despite the wide range of temperatures in which octopuses live. However, the question remains whether the adaptive editing change is due to environmental factors associated with the different water temperatures, or due to a genetic change that affects RNA editing in *cis*.

*Drosophila* is a useful model system to investigate the influence of environmental and genetic changes on RNA editing levels. Previous work in our lab and others shows the importance of genetic *cis*-regulation in comparing editing levels between different *Drosophila* species[13], and in *Drosophila* that come from a common environment[14,15], while other work in flies has shown that RNA editing changes in response to temperature[16,17]. Although examples of *cis*-regulation and *trans*-regulation of RNA editing have been identified, we do not know the relative importance of these modes of regulation when comparing flies from different environments.

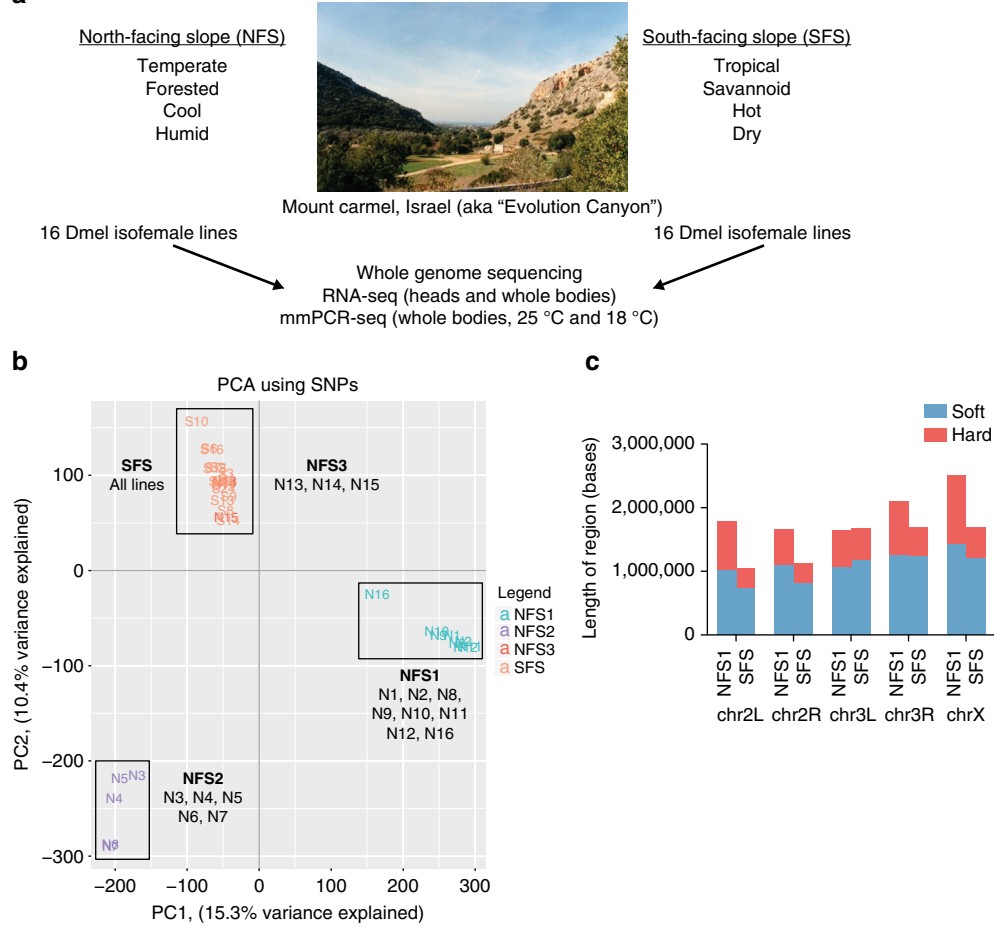

**Fig. 1** Genetic differences and selective sweeps in *D. melanogaster* from Evolution Canyon. **a** Schematic detailing the sequencing experiments performed. Photo credit: Michael Margulis. **b** PCA plot using SNPs from the 32 Evolution Canyon fly lines, with a single fly represented per line. The sub-populations are labeled as follows: NFS1 (turquoise), NFS2 (purple), NFS3 (red), SFS (orange). **c** Bar plot showing length of hard and soft sweep regions in 4 NFS1 and 12 SFS lines for different chromosomes

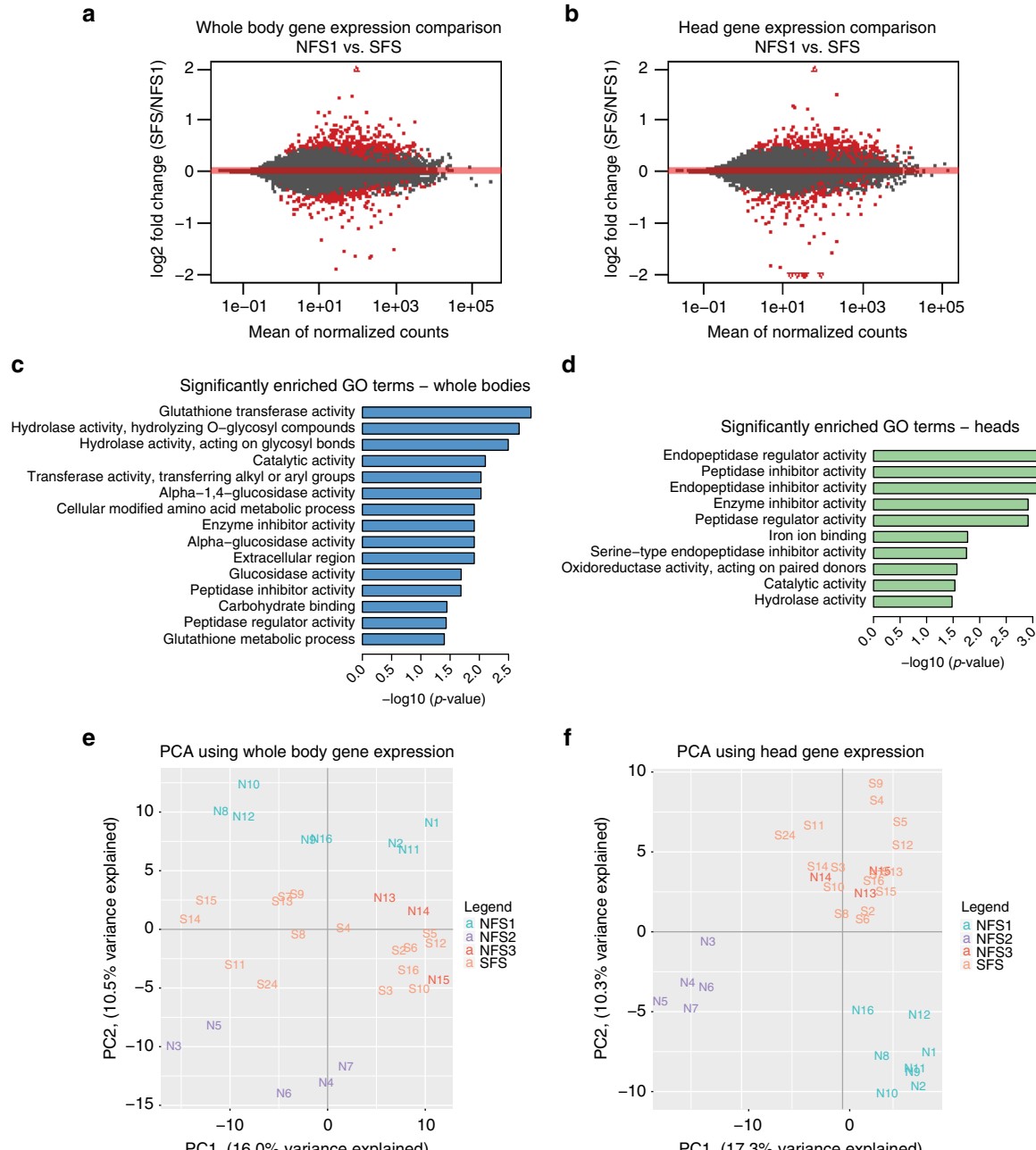

**Fig. 2** Gene expression differences between NFS1 and SFS populations reflect genetic differences. **a**, **b** Scatter plot showing gene expression differences between the 8 NFS1 and 16 SFS fly lines in whole bodies **a** and heads **b**. Red points represent genes with significantly different expression levels (Benjamini-Hochberg adjusted *p*-value <0.05, Wald test, DESeq2). **c**, **d** Significantly enriched GO-terms of genes over-expressed in SFS in whole bodies **c** and heads **d** (Benjamini-Hochberg adjusted *p*-value <0.05, GOseq using the Wallenius approximation). Abbreviated GO-terms for **c**, **d**: "transferase activity, transferring alkyl or aryl groups" represents "transferase activity, transferring alkyl or aryl (other than methyl) groups"; "oxidoreductase activity, acting on paired donors" represents "oxidoreductase activity, acting on paired donors, with incorporation or reduction of molecular oxygen". **e**, **f** PCA of whole body **e** and head **f** gene expression for all 32 Evolution Canyon fly lines, with 5 flies represented per line. The sub-populations are labeled as follows: NFS1 (turquoise), NFS2 (purple), NFS3 (red), SFS (orange)

To determine how genetics and environment affect gene expression and RNA editing, we examine *Drosophila melanogaster* flies that live along a microclimatic gradient in Evolution Canyon I (EC I, located in Lower Nahal Oren, Mt. Carmel, Israel). The north-facing slope (NFS) of the canyon is shaded, humid, and cool, while the south-facing slope (SFS) is sunny, hot, and dry[18]. Studying flies in Evolution Canyon presents a unique opportunity to identify genomic regions of selection that occur due to adaptation to different microclimates. Previous studies

have already demonstrated genetic differentiation and patterns of local adaptations in *D. melanogaster* from the two abutting slopes[19,20], although gene expression and RNA editing have not yet been systematically examined in these flies. We perform whole-genome sequencing (WGS) and RNA-seq on individual fly lines from Evolution Canyon to investigate how DNA mutations modulate differences in gene expression in flies between the NFS and SFS. We also measure RNA editing levels in these lines to determine the role of genetic *cis*-regulation in guiding editing

level differences, and how this mode of regulation compares and interacts with environmental regulation of editing. With this study, we form a more complete picture about the various contributions to gene expression and RNA editing differences in flies from Evolution Canyon, results that can potentially be applied to other species and climates.

## Results

**Inter-slope and intra-slope genomic divergence and differentiation.** To determine the extent to which genetic *cis*-regulation affects gene expression and editing levels in flies from the opposite slopes of Evolution Canyon, we sequenced individual flies from 16 isofemale lines from the south-facing slope (SFS) and 16 isofemale lines from the north-facing slope (NFS) (Fig. 1a). Specifically, for each of these 32 lines, we performed WGS, RNA-seq of whole body and head tissue of flies raised at 25 °C, and mmPCR-seq of whole body tissue from flies raised at 25 and 18 °C. mmPCR-seq is a targeted sequencing method our lab recently developed that allows us to measure editing levels at high resolution for many samples simultaneously[21].

Since the sample collection sites of the two slopes of the canyon are only separated by 250 meters on average, and inter-slope migration of the flies has been previously recorded[22], we first wanted to determine if any of the NFS or SFS flies were migrants from the opposite slope that they were collected from. To do this, we performed principal component analysis (PCA) using data from ~2 million SNPs identified from WGS (Fig. 1b). While the SFS lines all form a single cluster, the NFS lines formed three different groups, which we label 'NFS1', 'NFS2', and 'NFS3'. The NFS3 group consists of three isofemale lines that are in the same cluster as the SFS lines, suggesting that these flies are migrants from the SFS. Much higher migration rates were recorded from SFS to NFS than the reverse[22], so the presence of some NFS flies among the SFS flies is expected. Still, the presence of the other two distinct PCA clusters within the NFS implies overall higher diversification of this group compared with the SFS. Since the lines in the NFS1 group are separated from the rest of the flies on the PC1 axis, which explains the most variation in the data set (15.3%), and since the NFS1 group contains more flies than the NFS2 group, we considered the flies in the NFS1 group to most accurately represent the NFS flies. However, the NFS2 lines may also be a relevant population that warrants further study. Thus, for completeness, we analyzed the 8 NFS1 lines and 5 NFS2 lines separately, with the main figures representing comparisons between the NFS1 and SFS lines, and those pertaining to the NFS2 and SFS lines in the supplementary figures. Since the SFS lines were relatively tightly clustered together, we used all 16 of these lines in subsequent analyses. For clarity, we will refer to these specific clusters as 'NFS1', 'NFS2', and 'SFS'.

Next, we wanted to determine which fraction of the standing genomic variation is under selection, and thus likely to be adaptive. To do this, we identified selective sweeps genome-wide. Selective sweeps are regions of DNA that show reduced nucleotide diversity, as a result of one or more beneficial alleles in that region reaching fixation in the population. Although selective sweep regions were previously identified in these Evolution Canyon fly lines, they were determined from genome sequencing of the pooled NFS and SFS lines[19]. We re-calculated the sweep regions with the new genome sequencing data from each individual fly line and also took into account the new fly group clustering from the PCA results (Methods section), enabling distinction between hard and soft sweeps. In a hard sweep, one newly occurring mutation is rapidly swept up to fixation in a population, while in a soft sweep, multiple independent mutations are simultaneously swept up in a

population[23]. Sweep regions with total length 9.71 Mb (with 5.85 Mb or 60% classified as soft sweeps and 3.86 Mb or 40% classified as hard sweeps) and 8.26 Mb (with 6.37 Mb or 77% classified as soft sweeps and 1.88 Mb or 23% classified as hard sweeps) were found in the NFS1 and NFS2 populations, respectively. Sweep regions of a smaller size (7.22 Mb) were identified in the SFS population, with 5.16 Mb (71%) classified as soft sweeps and 2.06 Mb (29%) classified as hard sweeps (Fig. 1c, Supplementary Fig. 1, Supplementary Data 1). A visualization of where these sweeps are located along each chromosome is presented in Supplementary Fig. 2, and demonstrates that the NFS1, NFS2, and SFS flies have experienced unique evolutionary changes.

**Comparison of differential gene expression and SNP patterns.** We next determined how gene expression differs between the NFS and SFS fly lines. We performed RNA-seq on head and whole body tissue from each fly line. We identified 515 and 449 genes that were differentially expressed between the NFS1 and SFS lines in the whole body and head tissue, respectively (Fig. 2a, b, Supplementary Data 2). Gene ontology (GO) analysis using GOseq[24] shows several significantly enriched GO-term groups (Fig. 2c, d), specifically in genes with higher expression in the SFS flies than the NFS1 flies. In whole bodies, glutathione transferase activity shows the highest enrichment. These genes make up a family of enzymes that are involved in detoxifying xenobiotics. Upon further examination, we found that genes with glutathione transferase activity are globally over-expressed in the SFS compared to NFS1 (Supplementary Fig. 3a). We also observed hydrolase activity acting on glycosyl bonds to be highly enriched in whole bodies, and peptidase inhibitor activity shows enrichment in both whole bodies and heads. In addition, we examined genes involved in pigmentation, DNA repair, and heat stress response. Based on previous studies, one might expect these genes to be more highly expressed in flies living in the SFS given its increased sunlight, radiation, and heat relative to the NFS[25–28]. For pigmentation, we observed all genes that were differentially expressed to have higher expression in the SFS flies (Supplementary Fig. 4a, b). In addition, we observed that *TotX*, part of the stress-inducible humoral factor *Turandot* genes[29], is over 10 times more expressed in whole flies from SFS compared to NFS1. Additional *Turandot* genes show higher expression in SFS flies as well (Supplementary Fig. 4c).

In comparing the NFS2 and SFS lines, we identified 612 and 599 differentially expressed genes in the whole body and head, respectively (Supplementary Fig. 5, Supplementary Data 2). We observed several significantly enriched GO-terms for these genes as well (Supplementary Fig. 6). For whole body tissue, we observed significantly enriched GO-terms for iron ion binding in genes that were over-expressed in the NFS2 fly lines. For head tissue, we observed several chitin and metabolism-related GO-terms to be significantly enriched in genes that were over-expressed in the NFS2 fly lines. Genes over-expressed in the SFS fly lines were significantly enriched with contractile fiber-related terms. Although glutathione transferase activity did not show significant GO-term enrichment, we observed that many of these genes were over-expressed in the SFS compared to the NFS2 fly lines, similar to the SFS and NFS1 lines (Supplementary Fig. 3b). Thus, while the NFS1 and NFS2 lines appear to have adapted to the NFS largely independently from each other, there appear to be some similarities between them.

Given that we see hundreds of genes differentially expressed between flies from the two slopes, we wanted to determine whether the patterns in gene expression differences reflected the genetic differences between the flies. We first performed PCA

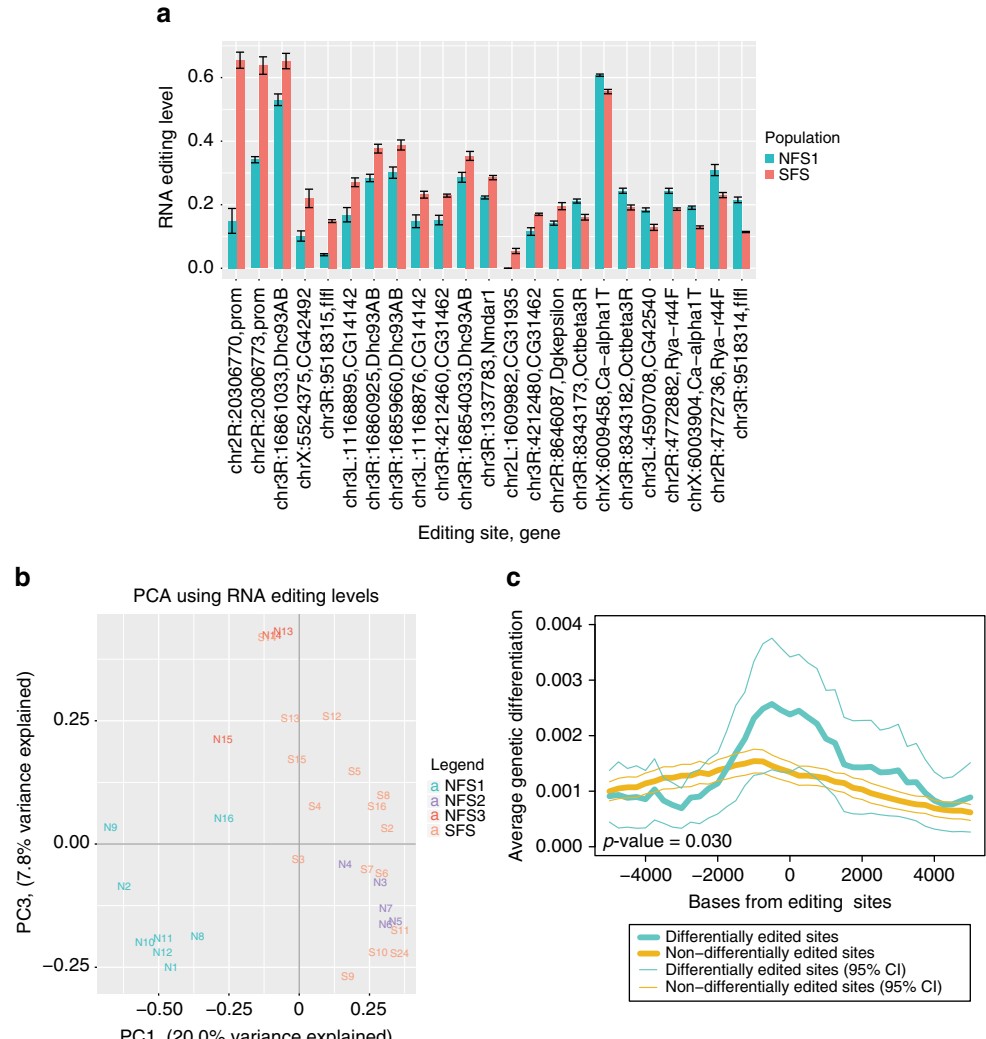

**Fig. 3** RNA editing differences between NFS1 and SFS populations reflect genetic differences. **a** Barplot showing the average editing levels of sites with significantly different RNA editing levels between NFS1 (turquoise) and SFS (red) flies, from mmPCR-seq (t-test, FDR-adjusted p-value <0.05, editing level difference ≥ 5%). Error bars represent standard error of the mean. The number of fly lines represented for each site per population ranges from 4 to 8 for NFS1 and 11 to 16 for SFS; Supplementary Data 6. **b** PCA plot using mmPCR-seq editing level measurements from all 32 Evolution Canyon fly lines, with average editing levels from two biological replicates represented per line. The sub-populations are labeled as follows: NFS1 (turquoise), NFS2 (purple), NFS3 (red), SFS (orange). PC2, which represented 16.1% of the variance explained, was excluded from the plot because it correlated with when the samples were sequenced (Supplementary Fig. 7b). **c** Plot showing significant enrichment of differentiated SNPs near significantly differentially edited sites (turquoise) compared to non-differentially edited sites (gold) between the NFS1 and SFS fly populations, along with the 95% confidence interval (p-value = 0.030, one-sided KS test)

using the gene expression levels of head and whole body tissue for all Evolution Canyon fly lines. A previous study which used PCA to compare genetic and gene expression differentiation between different human populations did not show similar groupings between the two data sets[30]. However, we observe clusters in the gene expression PCAs that mirror the clusters observed from the WGS PCA (Fig. 2e, f). In addition, when we compare the PC values from the SNP PCA (Fig. 1b) and PC values from these gene expression PCAs, we observe a significant correlation between PC1 of the SNP PCA and PC2 of the whole body gene expression PCA (Spearman rank $r = 0.810$, p-value $= 6 \times 10^{-7}$), as well as between PC1 of the SNP PCA and PC1 of the head gene expression PCA (Spearman rank $r = 0.664$, p-value $= 5 \times 10^{-5}$), suggesting that genetic differences between the fly groups are strongly associated with the gene expression differences. The clustering in the gene expression PCAs is not correlated with when the samples were sequenced (Supplementary Fig. 7a).

We then determined whether SNPs were enriched in the significantly differentially expressed genes. To do this, we calculated the Fixation Index, or Fst, of each SNP. The Fst measures the amount of genetic differentiation between different populations. We then determined the number of genes that had at least one highly differentiated SNP (top 0.5% Fst). Using the R package GOseq[24], which takes into account gene length, we found a significant enrichment of genes with at least one differentiated SNP in the genes with altered expression between the NFS1 and SFS flies in the head (78/424 (18.4%) vs. 1353/10900 (12.4%), p-value $= 2.6 \times 10^{-4}$), as well as in the whole body (89/493 (18.1%) vs. 1501/13394 (11.2%), p-value $= 1.26 \times 10^{-6}$) (see Methods section for details). In addition, when we examine potential promoter regions of genes (1 kb upstream), we observe a similar enrichment. For significantly differentially expressed head genes, we find 30/424 (7.1%) of these genes have at least one differentiated SNP in the upstream 1 kb region, whereas 491/10900 (4.5%) of non-significant genes do

($p$-value = 0.013, one-sided Fisher's exact test). For significantly differentially expressed whole body genes, we find 44/493 (8.9%) of these genes have at least one SNP in the upstream 1 kb region, whereas 637/13394 (4.8%) of non-significant genes do ($p$-value = $8.8 \times 10^{-5}$, one-sided Fisher's exact test). We observe similar results when examining significantly differentially expressed genes between the NFS2 and SFS fly lines (Supplementary Data 3). This provides evidence that genetic mutations within the gene of interest and in potential regulatory regions are likely contributing to gene expression differences between flies from the opposite slopes.

We next wanted to determine which genes were most likely to have potentially adaptive regulatory mutations responsible for gene expression differences between the NFS1 and SFS flies, and the NFS2 and SFS flies. To do this, we first identified the highly differentiated (top 0.5% Fst) SNPs and indels that overlapped with a selective sweep in the NFS1 or SFS flies, and the NFS2 or SFS flies (Supplementary Data 4). Then, we determined which of these mutations occurred in the region 1 kb upstream of the transcription start site, the 5′UTR, or 3′UTR of the whole body and head differentially expressed genes. For the NFS1 and SFS flies, we were left with 20 genes (Supplementary Data 5). The functions of these genes include DNA binding (*bowl* and *spn-D*), endopeptidase activity (*Tep2* and *CG6067*), and ATP binding (*CG31689*, *Gem3*, *spn-D*), among others. For the NFS2 and SFS flies, we were left with 17 genes, including those involved with DNA binding (*Nup62*, *twi*, *ovo*), and glutathione hydrolase activity (*Ggt-1*) (Supplementary Data 5). These genes and their associated mutations are candidates for future validation experiments of gene expression regulation and adaptation in Evolution Canyon flies.

**RNA editing level differences and genetic *cis*-regulation**. Next, we examined how editing levels differ between flies from the opposing slopes. We grew the flies at room temperature, collected two whole bodies (biological replicates) per line and measured editing levels of these flies using mmPCR-seq. mmPCR-seq is a microfluidics-based multiplex PCR method that allows us to efficiently measure RNA editing levels at much higher coverage than traditional RNA-seq, providing better resolution of the editing level measurements[21]. Overall, the biological replicates showed consistent editing levels (Supplementary Figs. 8, 9). After filtering out low-quality editing measurements (Methods section), we identified 51 editing sites that have significantly different editing levels between the NFS1 and SFS lines by a *t*-test (Supplementary Data 6). The sites that show the greatest differences in editing are shown in Fig. 3a. Several of the differentially edited sites generate non-synonymous amino acid changes and are in neuronal genes, which is typical of many editing sites in *Drosophila* (Supplementary Table 1). We also calculated differentially edited sites from the RNA-seq of fly head tissue, where we identified several differentially edited sites that were not covered in the mmPCR-seq assay, including non-synonymous, intronic, and intergenic sites. In total, we identified 9 differentially edited sites from the RNA-seq of fly head tissue (Supplementary Fig. 10, Supplementary Data 7), 7 of which were not covered in mmPCR-seq. We note that sites may be differentially edited because of a change in the relative expression of different splice variants that are edited at specific frequencies, rather than a change in editing within a particular splice variant. In comparing the NFS2 and SFS flies, we identified 34 editing sites through mmPCR-seq that have significantly different editing levels by a *t*-test (Supplementary Data 6); those sites showing at least a 5% difference in editing are presented in Supplementary Fig. 11, Supplementary Table 2. In analyzing the RNA-seq from head tissue, we identified 13 differentially edited sites between the NFS2 and SFS flies, 9 of which

were not covered in mmPCR-seq (Supplementary Fig. 12, Supplementary Data 7). Similar to the comparison between the NFS1 and SFS flies, we identified several non-synonymous differentially edited sites in neuronal genes from mmPCR-seq, with additional non-synonymous, intronic, and 3′ UTR sites from RNA-seq (Supplementary Table 2).

As with the gene expression data, we performed PCA using all of the Evolution Canyon fly lines to see whether RNA editing differences between the fly lines are associated with genetic differences. We observed that PC2 correlates with the time the samples were sequenced (Supplementary Fig. 7b), and that fly lines in the NFS1 group are separated out from the remaining fly lines along the PC1 axis (Fig. 3b); specifically, we see a significant correlation between PC1 of the SNP PCA (Fig. 1b) and PC1 of the editing PCA (Spearman rank r = −0.780, $p$-value = $10^{-6}$). This hints at a connection between genetic and RNA editing differences between flies from the opposite slopes. To investigate this further, we first examined whether there is a significant enrichment of SNPs near differentially edited sites, and in their predicted Editing Complementary Sequences, or ECSs. The ECS is part of the core double-stranded RNA (dsRNA) region required for editing, and is the sequence opposite the editing sites in the secondary structure. We specifically identified how many sites have at least one highly differentiated SNP (top 0.5% Fst) within 100 bases, and how many have at least one differentiated SNP in their respective computationally predicted ECSs[14,31]. We found that 3/26 (11.5%) of differentially edited sites between the NFS1 and SFS fly lines are near at least one highly differentiated SNP, whereas 2/378 (5.3%) of non-differentially edited sites are near one ($p$-value = 0.0022, one-sided Fisher's exact test). Similarly, we found that 3/26 (11.5%) of differentially edited sites have at least one highly differentiated SNP in their respective predicted ECSs, while none of the non-differentially edited sites have one ($p$-value = 0.00024, one-sided Fisher's exact test). For the comparison between the NFS2 and SFS lines, we found that 1/21 (4.8%) of differentially edited sites are near a highly differentiated SNP, while 1/382 (2.6%) of non-differentially edited sites are near one ($p$-value = 0.10, one-sided Fisher's exact test); however, we did not find highly differentiated SNPs in the predicted ECSs of either the differentially or non-differentially edited sites. Although we see a significant SNP enrichment for differentially edited sites between the NFS1 and SFS lines in both of these analyses, the absolute numbers of SNPs identified near differentially edited sites are relatively low; this may be due to the limitation of predicting ECSs, as well as the fact that SNPs in multiple locations are affecting editing levels. Thus, to further determine whether there is a significant enrichment of genetic differentiation around differentially edited sites, we quantified the amount of genetic differentiation around sites that are significantly differentially edited between the NFS1 and SFS lines, and separately around the non-differentially edited sites (Fig. 3c); we also performed this analysis for the NFS2 and SFS lines (Supplementary Fig. 13). For both sets of comparisons, we observed a significantly higher amount of genetic differentiation around the differentially edited sites compared to the non-differentially edited sites, which again provides evidence that genetic differences, and more specifically SNPs affecting local *cis*-regulation, are largely guiding editing level differences in flies from the abutting slopes.

**An intronic SNP modulates *prominin* editing levels**. To confirm the role of genetic *cis*-regulation in modulating editing level differences between the NFS1 and SFS flies, we zoomed in on two sites in the gene *prominin* that showed the largest changes in editing between the NFS1 and SFS slope populations (Figs. 3a, 4a). These sites are 3 bases away from each other and show similar

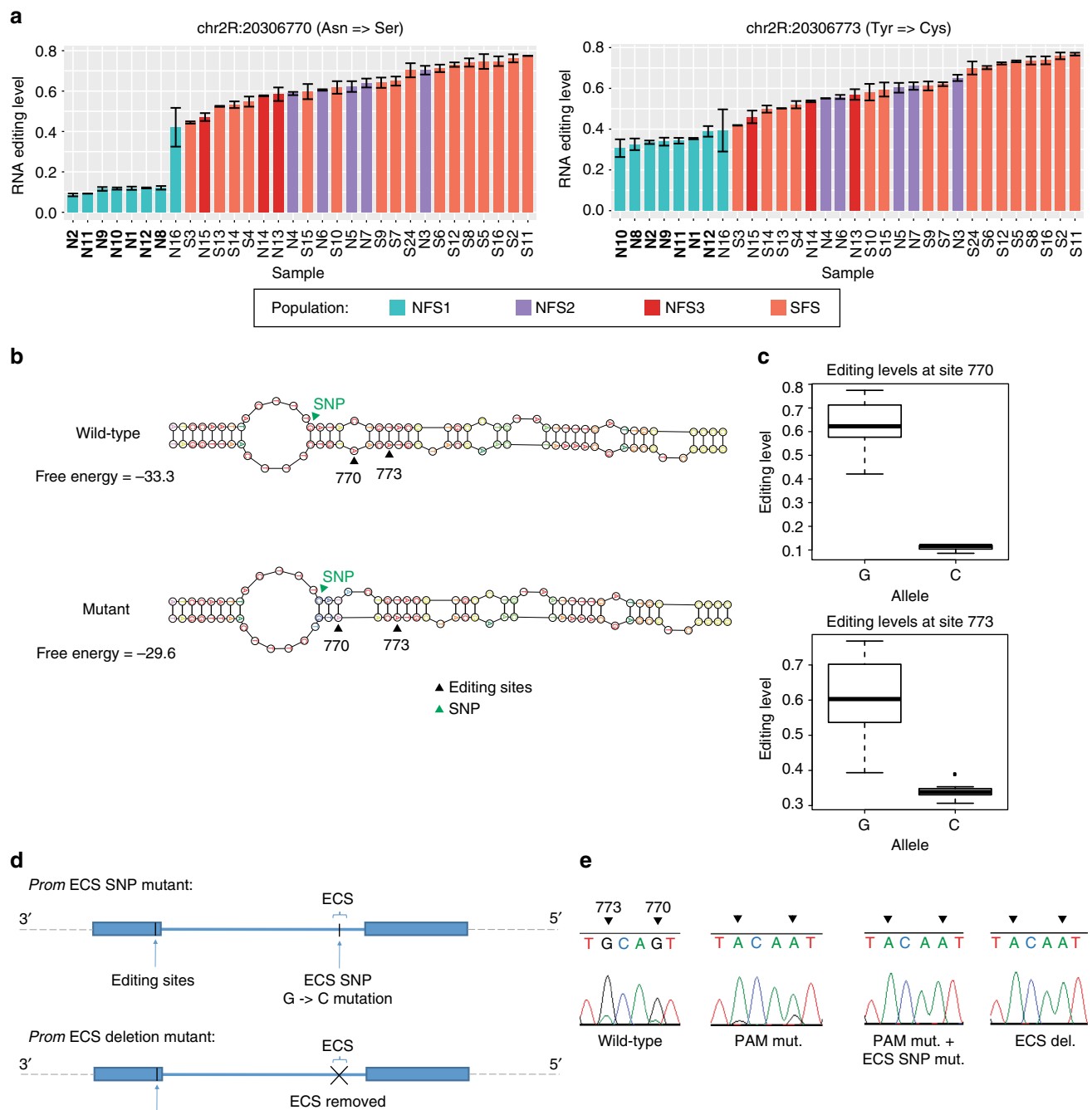

**Fig. 4** An intronic SNP affects *prominin* editing levels. **a** *Prominin* editing levels for each of the 32 Evolution Canyon fly lines. The editing levels for each sample represent the average editing levels of two biological replicates. The sub-populations are labeled as follows: NFS1 (turquoise), NFS2 (purple), NFS3 (red), SFS (orange). Samples in bold represent those that have the SNP that is correlated with *prominin* editing levels. Error bars represent standard error of the mean. **b** RNA structure predictions for the *prominin* editing sites and their corresponding ECS, for both wild-type *prominin* and with the SNP in the *prominin* ECS that is correlated with its editing levels. **c** Boxplot showing editing levels for *prominin* site 770 and site 773, for 7 Evolution Canyon fly lines with wild-type alleles and 25 Evolution Canyon fly lines with mutant alleles. The editing levels for both sites are represented by 7 NFS1 lines and 16 SFS lines. **d** Schematic showing the *prominin* ECS deletion and ECS SNP mutation. **e** Sanger sequencing traces showing *prominin* editing levels in whole bodies of wild-type, PAM mutant, PAM mutant with ECS SNP mutant, and ECS deletion mutant flies. Each trace represents editing levels from a single fly

trends in editing differences among the fly lines. *Prominin* is a membrane protein in photoreceptor cells that is involved with rhabdomere separation[32,33]. The two editing sites (which we will refer to as "770" and "773" based on their position in the genome) each encode a non-synonymous amino acid change (Asparagine to Serine for site 770 and Tyrosine to Cysteine for site 773), and are located in one of the extracellular domains of the protein[34,35].

Interestingly, the distribution pattern of the different editing isoforms also differed between the NFS1 and SFS flies (Supplementary Fig. 14). The sites are also highly conserved across different *Drosophila* species (Supplementary Fig. 15)[36].

We hypothesized that a SNP exists near these editing sites that is largely responsible for the editing level differences between the NFS1 and SFS1 groups. We observed several SNPs in the

*prominin* transcript that showed differentiation between the NFS1 and SFS flies (Supplementary Table 3). Intriguingly, one of these SNPs was located in the *prominin* ECS, and occurred in 7 out of the 8 NFS1 lines, but in none of the remaining Evolution Canyon lines. Although most SNPs that regulate editing levels in *cis* are close to editing sites, this SNP is located in an intron almost 1 kb away from the *prominin* editing sites. However, after

performing RNA structural prediction using RNAstructure software[37], this SNP is only 3 bases away from the editing sites in the secondary structure. Comparing the wild-type allele and mutant allele of the SNP, we see that the mutated SNP removes a base pair and alters the size and symmetry of two bulges near the editing sites (Fig. 4b). As a result, the predicted RNA structure containing the mutated SNP has higher free energy, and thus

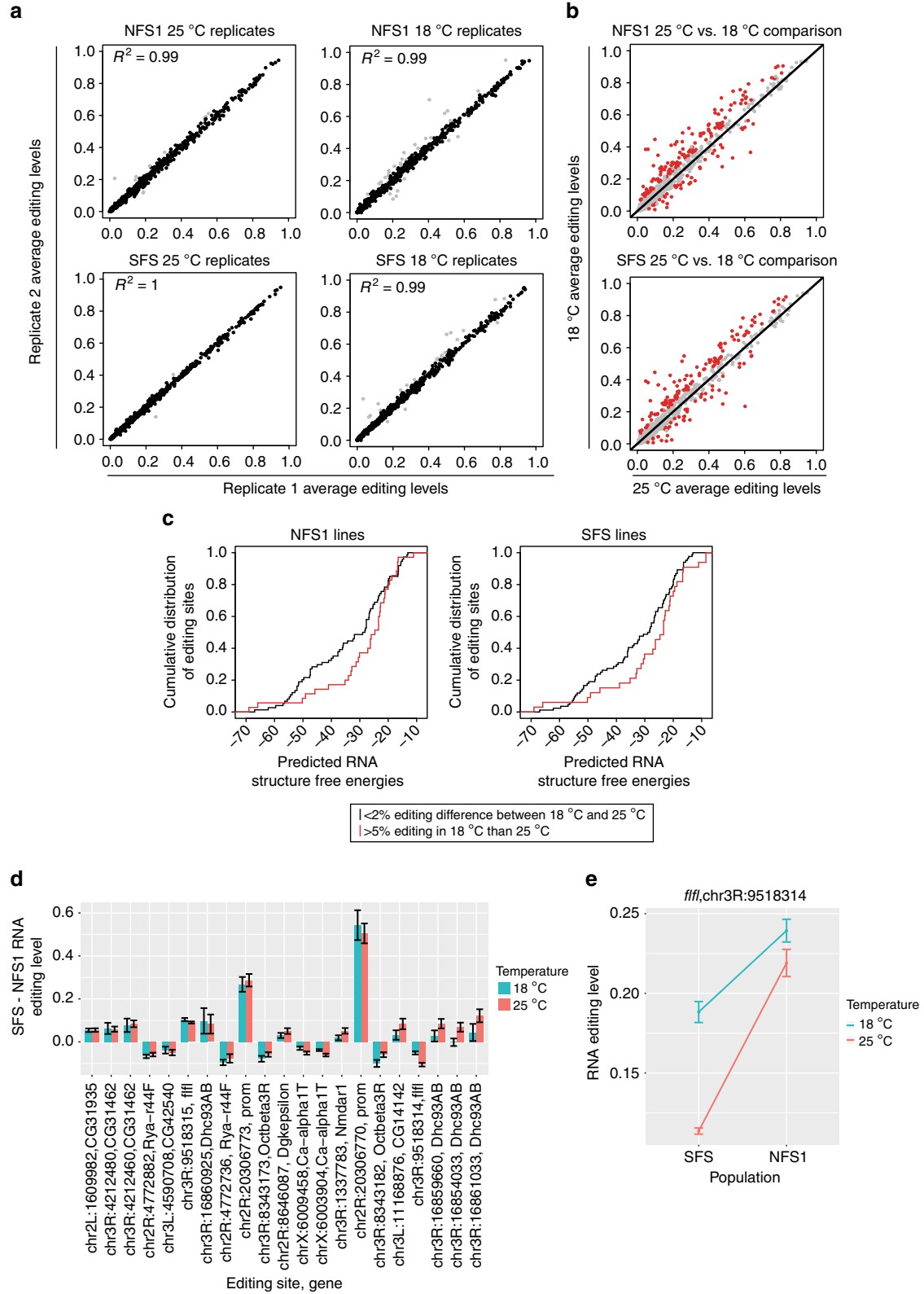

lower stability. In addition, the NFS1 lines with the mutated SNP have significantly lower *prominin* editing levels than the SFS lines (Fig. 4c). Since the SNP in the ECS is predicted to decrease the stability of the RNA structure, and the stability of the RNA editing substrate is correlated with editing efficiency, this suggests that this SNP is likely responsible for modulating *prominin* editing level differences between the NFS1 and SFS flies.

To confirm the effect of the SNP on editing levels, we used the gene editing tool CRISPR to generate different mutants that altered the *prominin* ECS sequence (Fig. 4d). By binding to a pre-designed guide RNA, the CRISPR-associated protein 9 (Cas9) endonuclease is directed to a specific region in the genome, where it can make precise genetic mutations[38]. We used the CRISPR-Cas9 system to make point mutations in the ECS, as well as remove the entire ECS sequence. In one mutant, which we call the *prom* ECS SNP mutant, the SNP in the ECS that we found to be correlated with *prominin* editing levels was mutated to the allele found in the majority of the NFS1 lines. We note that an additional nucleotide 10 bases away in the PAM sequence was also mutated to prevent repeated cutting and degradation by Cas9. To control for this change, we also generated a CRISPR mutant that only contained this PAM mutation, called the *prom* PAM mutant. In the other mutant, called the *prom* ECS deletion mutant, the ECS was removed entirely, as well as some of the flanking sequence. We measured editing levels of these flies using Sanger Sequencing and the ab1 Peak Reporter tool[39] (Fig. 4e). In the wild-type control flies, which have the same background as the mutants but did not generate the mutation of the interest, editing levels are 81% for site 770 and 79% for site 773, similar to the SFS fly lines. In the *prom* PAM mutant, editing levels decrease by a large amount: site 770 has 21% editing and site 773 has 8% editing. In the *prom* ECS SNP mutant, editing levels of both sites are not detected, and this is true for the *prom* ECS deletion mutant as well. The decrease in editing in both the *prom* PAM mutant and the *prom* ECS SNP mutant suggests that mutations throughout the ECS of *prominin* may affect its editing levels. Overall, this demonstrates that the *prominin* ECS is a key structural element needed to maintain high *prominin* editing levels, and that the SNP we identified in the ECS helps account for the difference in *prominin* editing levels between the NFS1 and SFS flies.

### Comparing genetic and environmental regulation of editing.
Lastly, we were interested to see how changes in environment modulate editing levels, and how this compares to the editing level changes we observed between the different fly populations. We compared editing levels measured using mmPCR-seq between Evolution Canyon flies raised at 25 °C and ~50% humidity, and flies raised at 18 °C and ~70% humidity. For simplicity, we will refer to these conditions as "25 °C and "18 °C",

respectively. When we compare average editing levels between replicates for NFS1, SFS, and NFS2 at both 25 and 18 °C, editing levels are very similar (Fig. 5a, Supplementary Fig. 16a). However, we see more frequent and larger changes in editing when we compare editing levels between flies raised at 18 and 25 °C, for both the NFS1 and SFS flies (Fig. 5b, Supplementary Data 8), as well as the NFS2 flies (Supplementary Fig. 16b, Supplementary Data 8). Of the 173 differentially edited sites (q-value <0.05 and >5% editing level difference) between 18 and 25 °C in the NFS1 flies, and the 166 differentially edited sites between 18 and 25 °C in the SFS flies, we found 137 sites that overlap. When these 137 sites are additionally overlapped with the 172 differentially edited sites between 18 and 25 °C in the NFS2 flies, 124 sites remain. Similar to previous reports[16,17], we observed that most sites show decreased editing at the elevated temperature. This decrease in editing is not due to decreased expression of *Adar* transcripts at 25 °C compared to 18 °C (Supplementary Fig. 17). This suggests that the changes in editing we observed may be largely due to changes in stability of the underlying RNA structure at different temperatures, which has been previously proposed[16,17]. One might expect that more stable RNA structures would be less affected by an increase in temperature, whereas less stable structures would be more likely to 'melt', or destabilize, even further. In this case, the sites that are in more stable structures should not show as large of a decrease in editing at the elevated temperature compared to sites in less stable structures. To test this hypothesis, we examined the distribution of predicted RNA structure free energies[14] for editing sites that showed small changes in editing between temperatures, and for sites which showed a large decrease in editing at 25 °C (Fig. 5c, Supplementary Fig. 18). When using the average editing levels of the NFS1 lines, we see that indeed, sites that show little change in editing between temperatures have significantly lower predicted free energies than sites that have decreased editing at 25 °C compared to 18 °C ($p$-value = 0.039, one-sided Kolmogorov−Smirnov test). We see a similar trend when examining average editing levels of the SFS lines ($p$-value = 0.094, one-sided Kolmogorov−Smirnov test), and the NFS2 lines ($p$-value = 0.054, one-sided Kolmogorov−Smirnov test). This demonstrates that the overall decrease in editing levels at 25 °C compared to 18 °C is possibly due to the destabilization of RNA structures at higher temperatures, rather than a decrease in the amount of *Adar* transcript.

Next, we wanted to examine how genetics and environment interact to modulate RNA editing levels. For sites we identified as significantly differentially edited and having at least a 5% difference in editing between NFS1 and SFS, we plotted the change in editing between the NFS1 and SFS flies at 25 °C and at 18 °C (Fig. 5d, Supplementary Data 9), and generated a similar plot for the differentially edited sites between the NFS2 and SFS flies (Supplementary Fig. 19, Supplementary Data 9). We notice

**Fig. 5** Environmental regulation of RNA editing in the NFS1 and SFS populations. **a** Scatterplots comparing average biological replicate 1 and replicate 2 mmPCR-seq editing levels for NFS1 and SFS samples at 25 and 18 °C. Gray points represent sites with >5% editing level differences between replicates. **b** Scatterplots comparing the average editing levels of NFS1 and SFS flies between 18 and 25 °C. Red dots represent sites with >5% editing level difference between the temperatures, and FDR-adjusted $p$-value <0.05 ($t$-test). The number of fly lines represented for each of these sites per population ranges from 3 to 8 for NFS1 and 3 to 16 for SFS; Supplementary Data 8. **c** For NFS1 and SFS flies, plots comparing predicted RNA structure free energy levels for sites that show <2% editing level difference between 18 and 25 °C (black), and sites that show >5% editing at 18 °C than 25 °C (red) ($p$-value = 0.0393 for NFS1 lines, $p$-value = 0.0949 for SFS lines, one-sided KS-test). **d** For differentially edited mmPCR-seq sites shown in Fig. 3a that had adequate coverage for samples at 18 °C (see Methods section for details), compare average editing level differences at 18 °C (turquoise) and 25 °C (red). Error bars represent standard deviation. The number of fly lines represented for each of these sites per population ranges from 4 to 8 for NFS1 and 10 to 16 for SFS; Supplementary Data 9. **e** Plot showing the editing levels of an editing site in *falafel* that shows an interaction between genetics and environment (FDR-adjusted $p$-value = 0.065, ANOVA test). Editing levels at 18 °C are shown in turquoise and at 25 °C are shown in red. Error bars represent standard error of the mean. Editing levels for the SFS fly population are represented by all 16 SFS lines, while those for the NFS1 population are represented by 7 and 8 NFS1 lines for 18 and 25 °C, respectively

that for most of these sites, the magnitude and direction of the change in editing are not significantly different between the two temperatures; that is, sites whose levels differ between populations tend not to be strongly affected by temperature changes. This suggests that while temperature does modulate editing levels, the genetic variants that are likely responsible for editing level differences between the flies from the two slopes have a dominant effect. However, we do observe one site in the 3′ UTR of the *falafel* transcript that shows a significant interaction between genetics and environment using a 10% FDR threshold in the NFS1 and SFS flies (Fig. 5e). *Falafel* is a regulatory protein that plays a role in the cell cycle, specifically in the asymmetrical division of neuroblast cells[40]. It has also been shown to be associated with DNA repair[41]. While editing levels of this site in *falafel* are relatively consistent between temperatures in the NFS1 flies, the levels diverge in the SFS flies. We identified two adjacent SNPs in the predicted ECS for this editing site that correlate with its editing level differences between NFS1 and SFS, one of which was previously identified as an editing-QTL for this site[14]. The mutant allele of this SNP forms an additional base pair in the predicted RNA structure[37], and occurs in the NFS1 lines which have higher editing levels, in accordance with the correlation between RNA structure stability and editing level (Supplementary Fig. 20). *Falafel* also happens to be in a selective sweep in the SFS lines (Supplementary Data 1), raising the possibility that this genotype-environment interaction is also important for adaptation. Interestingly, in comparing the NFS2 and SFS flies, we observed 10 editing sites with significant interactions between genetics and environment using a 5% FDR threshold (Supplementary Fig. 21, Supplementary Data 10). The majority of these sites are recoding and are in the gene *cacophony*, the main subunit of a voltage-gated calcium channel which is involved in several neurological processes, including neurotransmitter secretion at neuromuscular junctions[42] and male courtship behavior[43,44]. As some of these sites within the *cacophony* transcript are separated by more than 10 kb, it is likely that the regulation responsible for this genotype-environment interaction is more complex than that for the *falafel* editing site. Altogether, this analysis shows that while most differentially edited sites between the opposing slope populations show similar trends in editing regardless of whether the temperature is 18 or 25 °C, interactions between genetics and environment can occur between different fly populations with naturally occurring genetic variation.

## Discussion

We analyzed the genomes, transcriptomes, and editomes of individual fly lines from the two slopes of Evolution Canyon, building upon previous studies examining *Drosophila* from this canyon[19,20,45]. Evolution Canyon is uniquely suited for studying the biodiversity, evolution and adaptation of organisms that live in relatively close proximity to each other, and is a potential model for incipient sympatric speciation[45–48]. Although genetic and gene expression differences have been discovered previously in flies from different environments, most studies have examined fly populations at different latitudinal clines, and many signatures of adaptation found so far may be related to migration out of Africa. This is the first study to systematically examine gene expression and RNA editing differences in flies from different microclimates.

We identified several candidate genes whose expression may play an adaptive role in the Evolution Canyon flies. The most striking of these are the Glutathione S-transferase genes, which show global under-expression in the NFS1 flies compared to the SFS (Supplementary Fig. 3a), and tend to be under-expressed in the NFS2 flies compared to the SFS as well (Supplementary

Fig. 3b). These detoxification enzymes metabolize antioxidants, and their decreased expression in the NFS1 and NFS2 relative to the SFS flies may be related to the decreased sun exposure, as there is 200–800% more solar radiation on the SFS[18]. It is interesting to note that a previous study examining flies from Evolution Canyon showed enrichment of glutathione metabolism and transferase activity in genomic regions with evidence of inter-slope differentiating selection[19]. In addition, some Glutathione S-transferase genes have been shown to have significantly decreased expression in European flies compared to African flies in brain tissue[49]. Other genes involved in pigmentation, stress response, digestion, and chitin-related processes also showed significant gene expression differences between flies from the two slopes (Fig. 2c, d, Supplementary Figs. 4, 6). Future studies will be needed to address the potential adaptive role of the expression of these genes, and any regulatory mutations that are responsible for these gene expression changes.

We were able to show a strong connection between the genetic differences and the gene expression and RNA editing differences of the flies from the two slopes (Figs. 2e, f, 3b, c, Supplementary Fig. 13). This implies that, despite the flies being collected from Evolution Canyon years prior to our experiments, genetic regulation of gene expression and RNA editing still persists in the isofemale lines. In particular, we confirmed that an intronic SNP regulates the editing levels of two sites in the *prominin* transcript, although we note that we could not determine the exact amount of editing level regulation contributed by the SNP, since a CRISPR-associated PAM mutation was made in the same mutant. Since both the editing sites and the intronic SNP are conserved in many *Drosophila* species, and since most of the NFS1 lines contain the mutant allele that causes a decrease in editing, it is possible that the less stressful NFS environment decreased the strength of selection against this mutation. Although we have not yet identified an editing-related *prominin* phenotype, previous studies examining RNA editing evolution in different *Drosophila* species have demonstrated evidence of selection, especially for conserved, non-synonymous sites[31,50–52]. As the sites in *prominin* studied here are likewise conserved and code for non-synonymous amino acid changes, it's still possible that they play an adaptive role.

For RNA editing, we found that both population genetic differences and environmental differences can regulate editing between flies from the two slopes, and that the regulation seems to act through changing the structural stability of the RNA editing substrate. A change in environment regulates editing to a large extent for dozens of sites, most likely by affecting the stability of many RNA structures simultaneously. In contrast, genetic regulation is more site-specific, likely due to particular SNPs nearby editing sites which change the stability of the RNA structure containing those sites. This result is also supported by previous studies that examined editing level differences between and within *Drosophila* species[13,14,50]. Population genetic differences in editing tend to be maintained regardless of the environment in which we measured them, suggesting that genetic regulation may be more influential than environmental regulation of these sites. We also observed one 3′ UTR site in the *falafel* transcript that exhibits a genotype-environment interaction between the NFS1 and SFS fly populations (Fig. 5e), as well as several non-synonymous sites in the *cacophony* transcript between the NFS2 and SFS fly populations (Supplementary Fig. 21). Future studies will be needed in different populations and environments to determine whether these trends in editing happen universally.

To conclude, our study found surprising connections between genetics, gene expression, and RNA editing in flies from the distinct microclimates of Evolution Canyon. By sequencing

individual lines, we are able to show a clear correspondence between genotype and gene expression differences between flies from the two opposing slopes, some of which may be important for adaptation. In addition, we observe both genetic and environmental regulation of RNA editing in these flies, though the two modes of regulation seem to operate mostly independently of each other. This study sets the stage for future examinations of the regulation of adaptive gene expression and RNA editing differences, not only in other fly populations, but in other species as well.

## Methods

**Evolution canyon fly collection.** The 16 NFS and 16 SFS isofemale lines were established from single females collected from opposite slopes of Nahal Oren canyon in Israel in October 2010. For the WGS, we sequenced a single-male fly from each of the 32 Evolution Canyon isofemale lines. For the RNA-seq experiments, we pooled together either whole bodies or heads of five 3–5-day-old male flies from each of the 32 isofemale lines, raised at 25 °C. For the mmPCR-seq experiments, we collected two single 3–5-day-old male whole bodies from each of the 32 isofemale lines, which represent biological replicates. We collected flies raised at 25 °C and at ~50% humidity, and at 18 °C and ~70% humidity. Fly heads or whole bodies were flash frozen in liquid nitrogen at the time of collection for all experiments. All flies were raised on a standard molasses diet (Stanford University Fly Media Center) and on a 12:12 light/dark cycle.

**Sequencing library preparation.** For the WGS, fly DNA was extracted using the Qiagen DNeasy Blood and Tissue Kit. We used the Nextera DNA Sample Prep Kit to prepare the WGS libraries. Seven of the isofemale line samples (N1, N5, N10, N12, S2, S10, and S11) were sequenced as 76 bp paired-end reads on the Illumina NextSeq platform. The remaining 25 samples were sequenced as 150 bp paired-end reads on the Illumina HiSeq platform.

For the RNA-sequencing experiments, total RNA was extracted using the Agencourt RNAdvance Tissue Kit, followed by rRNA depletion and TURBO DNAse treatment (Life Technologies). The RNA-seq library was prepared using the KAPA Stranded mRNA-seq kit (Kapa Biosystems). The libraries were sequenced as 76 bp paired-end reads using the Illumina NextSeq platform.

For the mmPCR-seq experiments, total RNA was extracted using the Agencourt RNAdvance Tissue Kit, followed by DNAse I treatment (Thermo Fisher Scientific). cDNA synthesis was performed using the iScript Advanced cDNA synthesis kit (BioRad). The cDNA was pre-amplified using pooled primers designed to amplify 605 loci containing editing sites, followed by multiplex PCR using the Fluidigm Access Array system, following previously published protocols[14,21]. For flies raised at 25 °C, the amplicons were sequenced as 150 or 151 bp single-end reads on the Illumina NextSeq platform, except for the amplicons for sample S14, which were sequenced as 150 bp single-end reads on the Illumina MiSeq platform. For flies raised at 18 °C, the amplicons were sequenced as 76 bp paired-end reads on the Illumina NextSeq platform.

**Sequence mapping.** For the WGS reads, the first 6 bases of the reads were trimmed, then mapped using the Burrows–wheeler algorithm (BWA)[53] against the *D. melanogaster* genome (BDGP R5/dm3, Apr. 2006)[54,55] using the default settings. Duplicate reads were marked using Picard (http://broadinstitute.github.io/picard/), and duplicate and unmapped reads were removed using Samtools[56], as well as reads with a mapping quality less than 10. DNA variants were called using GATK[57–59].

The RNA-seq reads were mapped differently for profiling gene expression and for measuring RNA editing levels. For gene expression, the last 2 bases of the reads were trimmed and then mapped using Tophat[60,61], using the settings "-N 5 --segment-mismatches 3 --read-edit-dist 5". We used the *D. melanogaster* genome (BDGP R5/dm3, Apr. 2006)[54,55] and version 5.53 *D. melanogaster* Flybase gene annotations[62] to map the reads. For RNA editing, the last 2 bases of the reads were trimmed and then mapped using BWA[53] to the dm3 reference genome, along with 70 bp exonic sequences covering known splice junctions, which were obtained from the UCSC genome browser[36]. Duplicate reads were marked using Picard (http://broadinstitute.github.io/picard/), and duplicate and unmapped reads were removed using Samtools[56], as well as reads with a mapping quality <10.

The mmPCR-seq data from the flies raised at 25 °C was mapped separately for analyses where comparisons were made within the flies raised at 25 °C, and for comparisons between flies raised at 25 and 18 °C. For the former, we trimmed the first 19 bases which contain the mmPCR primer sequence, and trimmed after the 143rd base of the read. We mapped the reads as single-end using the samse command in BWA[53], allowing for a maximum of 6 mismatches. We mapped these reads to the dm3 reference genome, along with 120 bp exonic sequences covering known splice junctions, which were obtained from gene annotations from the UCSC genome browser[36]. For the latter, we trimmed the first 19 bases which contain the mmPCR primer sequence, and trimmed after the 75th base of the read. We mapped the reads as single-end using the samse command in BWA[53], allowing

for a maximum of 6 mismatches. We mapped these reads to the dm3 reference genome, along with 52 bp exonic sequences covering known splice junctions, which were obtained from gene annotations from the UCSC genome browser[36].

For the mmPCR-seq reads from the flies at 18 °C, we trimmed the first 19 bases which contain the mmPCR primer sequence, as well as the last base of the read. We used the BWA samse command[53], allowing for a maximum of 6 mismatches, to map these reads as single-end to the dm3 reference genome, along with 52 bp exonic sequences covering known splice junctions, obtained from gene annotations from the UCSC genome browser[36]. We did not include reads mapped to regions with editing sites that were covered by primers.

**Principal component analysis (PCA).** We used the prcomp function in R to perform the PCA. For the whole-genome sequencing PCA, we included SNPs that were called by GATK[57–59] and were covered by at least 5 reads. For the gene expression PCA, counts for all expressed genes were obtained using featureCounts from Subread[63,64], then were log2 transformed using DESeq2[65], and this served as input for the PCA. For the RNA editing PCA, we included editing levels from sites in the mmPCR-seq data, in which all 32 Evolution Canyon samples had at least 50 reads coverage, and showed <20% difference between biological replicates. We also removed sites that all had >98% editing or <2% editing in all samples.

**Identification of putative selective sweeps.** To minimize the possible sampling effect of related individuals, the 32 samples were first filtered based on their IBD (identical-by-descent) scores (pi_hat) until no IBD scores between two samples were >0.25. Identical-by-descent is a region of DNA that is shared between two or more individuals and is inherited from a common ancestor, and was computed using PLINK's identity-by-descent function[66]. This left 6 NFS samples (N1, N3, N6, N9, N12, and N16) and 12 SFS samples (S4, S7, S8, S9, S10, S11, S12, S13, S14, S15, S16, and S24). The 6 NFS samples were classified as NFS1 (N1, N9, N12, and N16) and NFS2 (N3 and N6) according to the PCA clusters. Two methods were used to detect the possible sweep regions within population. (1) Pool-hmm[67]: Sequencing reads within population were merged together and then mapped to the reference genome (dm3) using BWA[53] with default parameters. The parameters in Pool-hmm were set to be "-n 100 -c 5 -C 400 -q 20 -p -k 0.0000000001", and "–theta" was set to be the θ estimated for each population. (2) S/HIC[68]: It is a machine learning based method and capable of detecting soft and hard sweeps. Genotype for each sample was generated using GATK[57,58] and the haplotype for each sample was inferred using BEAGLE (version 4)[69]. A window size of 20 Kb was set while training and classifying by S/HIC. Sweep regions identified by the two methods were combined and only overlapping regions between the two were kept. Hard and soft sweep categories from S/HIC were applied.

**Differential gene expression analyses.** Counts for each gene were obtained using featureCounts from Subread[63,64], which were used in DESeq2[65] to identify differentially expressed genes (adjusted *p*-value <0.05, Benjamini-Hochberg correction).

For identifying differentially expressed genes involved in pigmentation, DNA repair, and heat stress response, we used the following respective GO-terms: GO:0043473, GO:0006974, and GO:0009408.

**SNP enrichment in differentially expressed genes.** To determine whether significant differentially expressed genes were enriched in SNPs, we first filtered out lowly expressed genes from the non-significant genes; specifically, we only included genes which had high enough expression to generate a *p*-value of differential expression from DESeq2, which is determined by an algorithm in the DESeq2 software. We only considered genes with annotations, which were from RefSeq[70], taken from the UCSC genome browser[36]. If a gene had multiple isoforms, the longest one was used for identifying whether the gene had a SNP or not. *P*-values were generated using GOseq[24], using a probability weighting function based on gene lengths, and the Wallenius approximation.

**SNPs and indels in differentially expressed genes in sweeps.** We identified the top 0.5% most differentiated SNPs and indels that were located in the 3′ UTR, 5′ UTR or 1 kb upstream of the transcription start site of significantly differentially expressed genes (both between NFS1 and SFS, and NFS2 and SFS), and were also located in a selective sweep that was unique to a specific population. We only considered genes with annotations, which were from RefSeq[70], taken from the UCSC genome browser[36]. If the 3′ UTR or 5′ UTR had different isoform annotations, the longest one was used for identifying whether those regions had a mutation or not; in addition, the 1 kb region was considered upstream from the 5′ UTR with the longest annotation. Molecular functions of the genes containing the SNPs and indels were taken from FlyBase[62].

**RNA editing comparison using mmPCR-seq data.** We measured editing levels at sites listed in the RADAR database[71]. Editing levels of a site were calculated by determining the fraction of reads with a 'G' nucleotide at that site. For each fly line, editing levels of each site were averaged between the two biological replicates. For the comparisons between NFS1 and SFS at 25 °C and between NFS2 and SFS at 25 °

C, and for the NFS1, NFS2, and SFS 25 vs. 18 °C comparisons, we only compared sites that had at least 50 reads in both biological replicates and had less than 20% difference in editing between replicates in at least three samples in each group being compared. We removed sites that had <2% editing in all samples prior to testing as well, as well as sites that were identified as a SNP by GATK in the WGS data[57–59]. To identify significantly differentially edited sites, we performed a t-test using the average editing level measurements, and using the p-values from the t-test, subsequently calculated q-values using the R qvalue package[72] to take into account multiple hypothesis testing. Sites with q-value <0.05 were labeled as significant. A site originally identified as significant between NFS1 and SFS at 25 °C (chr2L:3762337) was found to have discrepant editing level measurements between paired-end mapping vs. single-end mapping for certain samples at 18 °C. Therefore, this site was not included in mmPCR editing-related analyses. Molecular functions of genes containing significantly differentially edited sites were taken from FlyBase[62].

**RNA editing comparison using RNA-seq data.** We measured editing levels at sites listed in the RADAR database[71]. Editing levels of a site were measured by taking the number of 'G' nucleotide reads at the site and dividing by the total number of 'A' and 'G' nucleotide reads at the site. Only bases with a quality score of 20 or greater were considered in calculating editing levels. Since the coverage of editing sites was not as high in the RNA-seq compared to the mmPCR-seq data, we tested for significant differences in editing using a likelihood ratio test, which took into account coverage of the editing sites. Specifically, we use a beta-binomial model to test the differential editing level. At each site, the underlying editing level, $\pi$, in each strain is modeled as a beta distribution. Conventionally Beta(a, b) probability density function is written as equations (1) and (2):

$$f(\pi; a, b) = \frac{\pi^{a-1}(1-\pi)^{b-1}}{B(a, b)}, \tag{1}$$

where

$$B(a, b) = \frac{\Gamma(a)\Gamma(b)}{\Gamma(a+b)}. \tag{2}$$

We re-parametrize this density function by Beta($u, N$), such that $u = \frac{a}{a+b}$ and $N = a + b$. For strain i, conditioning on the total reads mapped to the site, $X_i$, the number of the edited reads, $X_i'$, is binomial: $X_i'|X_i \sim \text{Binom}(X_i, \pi_i)$. Marginally $X_i'$ has a beta-binomial distribution.

Under the null hypothesis of equal editing level, $\pi_i \sim \text{Beta}(u_0, N)$ for all strains; in other words, all $\pi_i$ are drawn from the same Beta distribution. Under the alternative hypothesis, we let $\pi_i \sim \text{Beta}(u_1, N)$ for the SFS flies and $\pi_i \sim \text{Beta}(u_2, N)$ for the NFS (NFS1 or NFS2) flies. The parameters, $(u_0, N)$ under the null hypothesis and $(u_1, u_2, N)$ under the alternative hypothesis are estimated by maximizing the joint likelihood given all observed RNA-seq editing data. We note that this model accounts for over-dispersion in the count data by allowing $\pi_i$ to vary between strains.

We included sites in which at least three samples in NSF1 or NFS2 and in SFS had a coverage of at least 5 reads. We removed sites that were identified as a SNP by GATK in the WGS data[57–59]. Using the p-values from this test, q-values were calculated using the R qvalue package[72]; sites with q-value <0.05 were labeled as significant. Molecular functions of genes containing significantly differentially edited sites were taken from FlyBase[62].

**RNA editing SNP enrichment analysis.** For all analyses examining SNP enrichment near, or in the ECS of differentially edited sites, the differentially edited sites contained significant sites from 25 °C whole body mmPCR-seq and head RNA-seq (q-value <0.05), that also had at least a 5% editing level difference between the NFS1 and SFS flies or between the NFS2 and SFS flies. The non-differentially edited sites included sites with q-value ≥0.05 from the 25 °C whole body mmPCR-seq. Since some sites are clustered together, which might confound our results, we also only included one representative site within each cluster for our analyses. Specifically, if editing sites were clustered together (within 100 bases for the analyses which counted the number of SNPs near editing sites and in the predicted ECSs of editing sites, and within 5 kb for the Fst enrichment plot and the accompanying one-sided KS-test), then either the editing site with the greatest editing level difference between the NFS1 and SFS flies, or between the NFS2 and SFS flies, was used (for the significantly differentially edited sites), or the editing site with the smallest editing level difference was chosen (for the non-differentially edited sites). For the analysis comparing average Fst for the differentially edited and non-differentially edited sites, we calculated an average Fst value at each base 5′ and 3′ to each editing site, going out to 5 kb from the editing site. To do this, we first summed the Fst value at each base for each editing site, summing separately for the differentially edited and non-differentially edited sites, and then divided by the respective number of sites. To test whether there was higher genetic differentiation around the differentially edited sites, we averaged this average Fst value over 250 bp non-overlapping windows, and performed a one-sided KS test. For the plots in Fig. 3c, Supplementary Fig. 13, the average Fst values are averaged over 1250 bp overlapping windows with a 250 bp step size.

**Generating *prominin* CRISPR mutants.** To generate the *prominin* editing CRISPR mutant lines, plasmids containing guide RNAs were injected into embryos of flies expressing Cas9[73] (BestGene, Chino Hills, CA, USA). The sequences inserted into the guide RNA plasmids are as follows: *Prom* ECS SNP mutation sense guide RNAs: 5′-TTCGCAAATTCGTTACTTCGACGC-3′, 5′-TTCGACGTA-CAGTATCAAAGTAAT-3′. *Prom* ECS deletion sense guide RNAs: 5′-TTCGCAGGAAAAATCGGTCTTTGG-3′, 5′-TTCGACGTGTGTATCCA-GATTCCC-3′. For the *prom* ECS deletion mutants, we used Cas9 to make two double-stranded DNA breaks around the ECS. These breaks were then repaired using non-homologous end joining. For the *prom* SNP mutation mutants, we used single-stranded oligos that had 60 bp flanking sequences on each side of the mutations for homologous recombination[74]. The sequences for these oligos are as follows: single-stranded oligo for ECS SNP mutant (ECS SNP and PAM mutation): 5′-CAGGTGAGTCCACGTCCACTAACCGCACGTGTGTATCCA-GATTCCCAGGTAGTCAAAGTTGAAGCCCAACATTCGAGCGTCGAAGTA ACGAATTTGGTTATCGAATTACTTTCATACTGTACGTGCCTGTAATTTT GATTGCCCCCAAAGACCGATTTTTCCTGCTCTCCGTGGGAAAAATGAA-CATAC-3′. Single-stranded oligo for control (PAM mutation only): 5′-CTCAGGTGAGTCCACGTCCACTAACCGCACGTGTGTATCCAGATT CCCAGGTAGTCAAAGTTGAAGCCCAACATTCGAGCGTCGAAGTAACGA ATTTGGT TATCGAATTACTTTGATACTGTACGTGCCTGTAATTTTGA TTGCCC CCAAAGACCGATTTTTCCTG CTCTCCGTGG-3′. Flies were crossed and Sanger-sequenced to ensure that they were homozygous for the mutation of interest.

**qPCR analysis.** For comparing *Adar* expression between samples at 25 and 18 °C, we used the following primers:
 Adar_F: 5′-GCCTCAGATACACGCGGATA-3′
 Adar_R: 5′-TGCCCGCTAATAC CTTTCGA-3′
 Rp49_F: 5′-CCGCTTCAAGGG ACAGTATC-3′
 Rp49_R: 5′-GACAATCTC CTTGCGCTTCT-3′
 We performed qPCR using the KAPA SYBR FAST qPCR kit (Kapa Biosystems), and the BioRad CFX96 qPCR system.

**Genotype-environment interaction analysis.** To determine whether any editing sites had a genotype-environment interaction, we fit a generalized linear model of the form G + E + G × E (where G represents NFS1 and SFS editing levels, or NFS2 and SFS editing levels, E represents 25 and 18 °C editing levels, and G × E is the interaction between them) for each editing site that passed the requirements stated in the "RNA editing comparison using mmPCR-seq data" Methods section, and used a Gaussian distribution to fit the model. An ANOVA test was then performed on the model, and q-values were generated from the p-values determined from the interaction term, using the R qvalue package[72].

**Data availability.** The WGS, RNA-seq, and mmPCR-seq data were deposited to Gene Expression Omnibus at the National Center for Biotechnical Information under the accession number GSE104085.

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

## Acknowledgements

We thank members of the Li lab for comments, as well as Hunter Fraser, Tom Clandinin, Joe Lipsick, and Stephen Montgomery for helpful discussions. We also thank Drew Stenesen and Helmut Kramer for assistance with phenotyping flies. This work was supported by an NIH Molecular Biophysics Predoctoral Research Training Grant T32 GM008294 (A.L.Y.), Binational USA-Israel Science Foundation Grant BSF # 2011438 (K. L., E.R., A.B.K., E.N.), NIH grant R21EY024125 (S.M. and A.C.Z.), Research Foundation of Ancell-Teicher for Genetics and Molecular Evolution (E.N.), and NIH grants R01 GM102484, R01 GM124215, and R01 MH115080 and the Ellison Medical Foundation (J. B.L.).

## Author contributions

A.L.Y., J.F., S.M., A.C.Z., and J.B.L. performed the experiments. A.L.Y., L.K., H.T., and P. M. performed computational and statistical analyses. K.L., E.R., A.B.K., and E.N. contributed fly lines. J.B.L. and E.N. supervised the study. A.L.Y. and J.B.L wrote the paper, with input from the other authors.

## Additional information

**Competing interests:** The authors declare no competing financial interests.

