## [Peer Review File · Nature Communications]

Reviewers' comments:

Reviewer #1 (Remarks to the Author):

Regulation of Gene Expression and RNA editing in *Drosophila* adapting to divergent microclimates at Evolution Canyon

Yablonovitch et al.

In this study the authors examine *Drosophila* from two microclimates within Evolution Canyon: the hot, dry South Facing Slope (SFS) and the cool, humid North Facing Slope (NFS). Whole-genome sequencing (WGS), RNA-seq, and microfluidic multiplex PCR-seq (mmPCR-seq) was performed on 16 isofemale lines from each microclimate. In the first set of experiments, the authors grew both populations of flies at 25°C and identified differences in genomic sequence, gene expression and RNA editing between the SFS and NFS flies. Approximately 500 transcripts were differentially expressed in both head and whole body samples. These gene expression differences could be partially explained by genetic differences between SFS and NFS flies: This is because there was an enrichment of SNPs in the promoters or coding sequence of differentially expressed genes compared to those genes with similar expression levels.

The authors then examined RNA editing in the two populations of flies and found ~60 differentially edited sites. Interestingly, the regions around these sites are also enriched for SNPs compared to sites that were not differentially edited. Previous studies indicate that editing levels can be modulated by mutations that modify the structure of the editing complementary sequence (ECS). Indeed, the authors convincingly show that mutations of the SNPs in ECS disrupt editing at two sites in one gene, *Prominin*. Finally the authors grow the two populations of flies under conditions that mimic the two microclimates, to test the impact of environment on RNA editing levels. They show that most of the editing differences observed between SFS and NFS flies at 25°C are not further exacerbated with temperature changes. (There is one exception to this statement: *falafel* shows temperature dependent RNA editing.) This suggests that the genetic variation between the two populations is responsible for the differences in editing, potentially differences in structure surrounding the differentially edited sites.

This study more generally addresses an interesting and outstanding question in the field of RNA editing: what is the mechanism behind the RNA editing differences observed in different populations of organisms (e.g. octopus populations @ different temps with differential editing)? The observation that the differential editing between SFS and NFS flies might be due to genetic changes within the ECS that occur during the divergent evolution of these two populations rather than a short-term or immediate environmental effect on editing is surprising and important. Unfortunately, the reader is distracted from this key result by the inclusion of less significant data. A shorter, more focused manuscript highlighting these editing findings would be stronger. In addition, the authors need to address the major concern below prior to publication.

Major Concern:

One of the key conclusions of the paper is based on two observations: 1) that differentially edited sites have genomic variation in the regions surrounding the site and 2) these variations or SNPs in the genomic sequence modify the editing frequency by changing the secondary structure of the ECS. The key figure supporting observation #1 (Fig. 3c) is hard to interpret. It is not clear from the methods what editing sites were included in this analysis: 60 differentially edited sites from whole body and head or just the 23 sites showing a >5% change in editing frequency? The y-axis of Fig. 3c shows tiny genetic differences of an aggregate plot. How many of the 60 differentially edited sites have one or more SNPs in the region surrounding a site, and how many have no SNPs. The authors have to present a more thorough and skeptical comparison between sites that are differentially edited and those that are not. The second key observation, i.e., that the genetic differentiation around the editing sites disrupted the ECS was only shown for the Prominin transcript. How many of the differentially edited sites have genetic changes that disrupt the predicted ECS? In other words, how many of the surrounding SNPs are in the ECS, i.e., how many of the 60 sites might be like the Prominin editing? Although the Prominin example is convincing, it is not clear that genetic variation in the ECS is the only or even the predominant mechanism driving differential editing.

Minor concerns:

1) The authors state that the changes in editing levels at higher temperatures are “not due to decreases in Adar transcript levels at 25°C”. Previous studies indicate that Adar protein levels decrease with temperature with dramatic differences between 20°C and 30°C (Reider et al., 2015). Does a change in Adar protein levels explain the editing differences?

2) On page 7 the authors state: “The RNA-seq data allowed us to identify several differentially edited sites that were not covered in the mm-Seq assay, including non-synonymous, intronic and intergenic sites”. This was confusing. Were these additional sites including in the analysis of differentially edited sites? Are they in addition to or included in the 51 sites from whole bodies and the 9 sites from heads?

3) How many editing sites show >5% difference when comparing SFS and NFS flies. Is it the 23 genes shown in Fig. 3a? Are these the same genes as in Fig. 5D (but there are 21 genes here). The figure legend of Fig. 5D says these are the same genes as Fig 2c?

Reviewer #2 (Remarks to the Author):

The topic of this study is trendy and interesting. Yablonovitch et al. want to detect SNPs with divergent allele trajectories between North and South populations (measured as F_{st}) and study whether candidate SNPs can explain divergent RNA editing and differential expression levels.

I have several important concerns about this study that would make me prevent this manuscript from publication.

Pp5, 114-126: The removal of NSF2 from the analysis is not justified. PC2 clearly separates NSF2 from SFS. The analysis should include NSF2 or at least two analyses should be done: one including NSF2 and other excluding it.

Pp5, 127-134: the authors mention a 9.71Mb sweep region, 5.85Mb of it as soft sweep region. Where is this region located? Why the author only mention this one when there are many more in suppl. Data 1? Where can I see this region in figure 1c? What is the point of all this paragraph?

Pp6, 148-151: "In addition, we examined genes involved in pigmentation, DNA repair and heat stress response, as one might expect these genes to be more highly expressed in flies lining in the SFS...". Why? These GO terms or related GO terms do not seem to have an enrichment of DE genes. Is this looks like an ad hoc analysis. If so, please justify why. In addition: What genes are those? How did the authors select them? According to a GO terms? If so, which GO terms? Please provide numbers.

Pp6, 156-164: I don't see any clustering in figure 2e. There is no way to split both populations in this figure. In addition, the authors say in pp5 – 123: "use only these 8 NFS lines in subsequent analyses", which is certainly not the case in figure 2c and others. Am I missing something? In addition, I don't understand the rationale in using PCA to prove that DE is due to genetic differentiation. If the two groups show DE at the same conditions (temperature) is because there is a genetic component. Why then do the authors need to do the PCA? In addition, what data is used for the PCA, DE genes or all expressed genes?

Pp6, 167: the authors should provide some summary statistics for the F_{st} . For instance, what is the rank of the top 0.5% F_{st} . In addition, I guess $\sim 2,000,000 \times 0.005 = 10,000$ is the number of candidate SNPs. How do the authors handle this data? In addition, what are these numbers: 77/428, etc. I guess 77 are DE genes that have at least 1 candidate SNP and 428 are DE with no candidate SNPs? If so, the authors must consider a bias due to gene length effect: i.e., the longer the gene the more likely it will have a candidate SNP just by chance. The authors may want to use similar approaches as those used to study enrichment of SNPs in GO terms.

Pp6, 171: "this provides evidence that genetic mutations within the gene of interest are contributing to gene expression differences...". So, if I properly understand this, this result implies that SNPs occurring at genes are likely involved in regulation of gene expression? To my understanding from other studies, most gene expression divergence within populations and between species is do to mutations/SNPs occurring in regulatory regions, not in the gene body.

Pp7, first paragraph: finding candidate SNPs in upstream regions of genes located at selected regions (sweeps) does not prove or indicates these are adaptive regulatory mutations. For instance: how many DE genes in these sweep blocks do not have candidate

SNPs. How many candidate SNPs are in -1kb of equally expressed genes located in these sweep blocks?

Pp7, last paragraph: This concern about PCA is similar as in Pp6, 156-164. It looks like there is some differentiation between NFS1 and south population in PC1, but not in PC2.

Pp8, first paragraph: it is difficult to evaluate figure 3c. 1) why do the authors need to normalize Fst? I think the authors should use the actual Fst. 2) higher Fst around significant sites does not prove a cis-effect in editing. 3) these lines represent averages. The authors should also plot the CI at 95%. KS test does not consider this variance.

Pp9, 258-260: this proves that ECS is involved in editing efficiency, but it does not prove that the alternative aimed SNP is adaptive.

Pp9-11, section "Comparing genetic and environmental regulation of editing in evolution...". If the authors want to study the genetic and environmental (temperature) components of RNA-editing variation they should fit the data to a linear model (LM, if the data is normally distributed) or to a generalized linear model (GLM, which is likely the correct one given the binary nature of the data – edited read v unedited read). A GLM will allow the authors to break the RNA editing differentiation into the different factors: G + T + GxT. I don't think there is any other alternative to this.

Pp15, 438: why do you filter sites with <20% difference between replicates? Idem for the >98% and <2%. You might be biasing your analysis.

Pp15, 443: what is identical-by-descent and how do you compute it?

Pp15, 451-457: what is a hard sweep and what is a soft sweep.

Pp16, Section "SNP enrichment in differentially expressed genes": "... we first filter out lowly-expressed genes from non-significant genes". Why? Do you also filter out lowly-expressed genes from significant genes? I'd strongly suggest doing such a filter before the DE analysis so that lowly-expressed or unexpressed genes are not included in the analysis (see edgeR manual). This is important as the FDR value depends on the number of tests. "we only included genes which had high enough expression to generate a p-value..." What is high enough? "We excluded genes non-significant genes whose length was shorter than the minimum significant gene length or longer than the maximum significant gene length" What is minimum and maximum significant gene lengths?

Pp16, 478: again, the authors are manipulating the data with any reason to do this. This might bias the analysis significantly and the conclusions.

Pp16, 480-483: the authors use a t-test. Here, and in the whole manuscript, the authors must justify that the t-test is the appropriate statistical framework. For instance, is editing level normally distributed? In addition, the authors apply Storey and Tibshirani's adjustment method for multiple testing (qvalue) and afterwards apply BH correction. This makes no

sense.

Pp16, 487: "we tested for significant differences in editing using a likelihood ratio test". How are the two likelihoods needed for the LRT computed?

Pp17, 497-500: "To test whether there was higher SNP enrichment for the significant sites, we averaged this normalized F_{st} value over 250bp non-overlapping windows and performed a one-sided KS-test". This DOES NOT prove that you have more SNPs, only proves that you have different F_{st} for the SNPs.

MINOR CONCERNS

Pp5, 127: the authors should explain what is a selective sweep, what is results is expected (reduction of nucleotide diversity, etc), and what is the difference between hard and soft sweeps.

Pp7, 157: what is the RNA-Seq data set?

Pp7, 175: "after overlapping these genes..." How many gene are "these genes"? How many do you expect to find?

Pp7, 196: how much is "several"?

Pp8, 230-231: I don't understand this sentence.

Pp8, 234 and on: MUTANT allele, instead of mutated allele? In addition, can you maybe provide a name for the mutant and the alleles. Otherwise is very confusing.

Pp15, 434: what is the genome coverage. This will help to evaluated the minimum cut of 5 reads.

Reviewer #3 (Remarks to the Author):

The ability to recode has been one of the most extensively studied aspects of RNA editing. Most of the seminal studies focused on how individual recoding events affected protein function. A basic question regarding recoding is why would an organism choose to alter RNA instead of DNA, the genomic blueprint? The obvious, and generally assumed answer is that RNA-level changes provide flexibility: RNA edits can be turned on or off, generating distinct protein isoforms. It has been hypothesized that this plasticity might be in response to environmental conditions, but few studies have addressed this directly. Yablonovitch et al add important data to this theme. Past studies have shown that RNA editing correlates with environmental variation. Using transcriptome wide screens of editing sites, this paper looks at plastic sites between drosophila populations and asks which are due to acclimation and

which are due to adaptation. Further, it asks how the adaptive sights might be generated. I think that the paper makes important contributions. However, there are some points that need to be addressed to strengthen conclusions and areas where the paper could be presented more clearly. The paper should be interesting to a broad audience.

Statistical analysis is appropriate

-Joshua Rosenthal, MBL

Major points:

1. At several places, the authors say that environmental differences act on editing in a more "global" manner. I'm not sure exactly what this means. The data in Fig 5b shows that while more sites are edited at a higher level at 18C, there are still plenty of examples of sites that are more highly edited at 25C. This is an important point because it's a central theme of the paper. Maybe I'm missing something, but I just don't see the changes as being global. In addition, it would be good to see how much overlap there is between the sites that significantly change with temperature between the NSF1 and SFS flies.

2. It would be good to have two more comparisons: how does this data compare with that presented in Reider et al.? Are the same sites changing with temperature? Also, at 25C, how well do the RNA editing levels (and sites) match between the evolution canyon strains and past data published for a lab strain (like Canton S)? It would be interesting to see how much plasticity there is in the editome and the authors have a good opportunity to show this.

3. Regarding the structure driving RNA editing levels of prominin sites 770 and 773, the authors state "that the SNP we identified in the ECS helps account for the difference in prominin editing levels between the NFS1 and SFS flies." As the authors are aware, the additional mutation that was engineered to abolish the unwanted PAM site complicates matters because this mutation alone leads to a drastic reduction in editing (81% > 21% at 770 and 79% > 8% at 773). By adding the SNP on top of this editing is abolished. The authors conclude that this further reduction in editing by adding the SNP shows that it can reduce editing on its own. It is also possible that it only can reduce editing when in combination with the PAM mutation. It would be good if there were some independent verification that the SNP alone can reduce editing

Minor points:

1. First sentence of abstract should be modified from "Determining the mechanisms by which a species adapts to its unique environment is a key endeavor in evolution, ecology, and conservation" to "Determining the mechanisms by which a species adapts to its unique environment is a key endeavor in THE STUDY of evolution, ecology, and conservation."

2. "Inosine gets recognized as guanosine during translation, so A-to-I RNA editing has the potential to alter amino acid sequences of proteins and affect other transcriptional processes⁶." A review shouldn't be referenced here. The authors should reference the

primary articles that support these themes.

3. The discussion of selective sweeps needs to be clarified and expanded. As it stands, it comes out of the blue. Many the readers of this paper will not be evolutionary biologists. For example, people from different areas of the RNA editing field will be interested in the findings and they may not be well versed in the concept of a selective sweep. A little more background and context is necessary.

4. Rob Reenan's paper about the guided evolution of the *Drosophila* synaptotagmin site across insects (Nature. 2005;434 :409-13) should be added to the discussion at several places.

5. For p5-6 when talking about GO term enrichment, make it clearer whether you're talking about enrichment in NFS or SFS.

6. When introducing the data on the "CRISPR mutants" a little general context is required. First of all, the name "CRISPR mutant" should be changed to something else. In addition, a couple sentences should be added explaining how the technique works and what it will do in this instance.

7. A couple sentences should be added to more clearly explain what "felafel" does.

8. Figure 5d is unclear. I understand that you are looking at sites that showed a difference between the SFS and NFS1 populations in figure 2. Then you acutely changed the temperature between 18C and 25C and looked for changes. But is this data from NFS1 flies or SFS flies?

8. It should be pointed out that the quantification of editing using RNAseq data cannot accurately deconvolute editing from splicing. That's to say, if different splice variants are edited at specific frequencies and their relative expression changes, this looks very much like changes in editing frequency within a splice variant. This should be mentioned as a caveat.

9. "Evolution Canyon is uniquely suited for studying biodiversity, evolution and 318 adaptation,"...this is propbably an overstatement.

Reviewer #4 (Remarks to the Author):

RNA editing contributes to the diversity of the *drosophila* proteome, to an extent far greater than in mammals, as was recently demonstrated by a number of groups. This post-transcriptional mechanism is well-suited for adaptation, as was demonstrated by Garret and Rosenthal (Ref. 10). However, the extent of adaptation through editing, and the mechanisms through which it occurs (genetic mutations vs. regulation) are still poorly understood. The present manuscript by Yablonovitch et al is therefore an important and timely contribution.

Overall, the work is comprehensive, insightful, and well-written, and I find it suitable for publication in Nature Communications. However, I have a few questions, comments and

suggestions that the author might want to consider prior to publication.

1. In p.5 (and support data 1) sweep regions are reported. Can one predict which newly fixated mutations can be associated with these regions, or at least report the mutations in these regions that distinct NFS and SFS lines?
2. The meaning of F_{st} (presumably, the fixation index) should be clearly defined.
3. The highly differentiated SNPs are defined as having 0.5% top F_{st} . Is the F_{st} calculated on a single site? If so, does this criterion mean that only one allele is present in one of the groups? How do the enrichment numbers look if one considers only SNPs in which one allele is seen in all of group A flies and the other allele in all of group B flies (perfect separation)?
4. What about the 9 genes in sweep regions – pls add to Supp table 1 the information on how many WT and mutant alleles were seen in each group (NFS1 and SFS). Most importantly, please specify in figure 4a which allele was observed in each of the flies at the SNP that seems to determine the editing level.
5. P.7 lines 195-198: It seems the order of the sentences should be reversed: "In addition, the RNA-seq data ... and intergenic sites. We identified ..."
6. Regarding sites 770 and 773 – as these are adjacent, it is likely that the mmPCR data should allow examination of the distribution of the four possible pairs of aa's (NY, NC, SY, SC). Comparing this distribution between the groups may provide more information and increased statistical significance.
7. P.8 line 231 – please expand and provide evidence supporting the claim that most SNPs regulating editing are in cis. If this is correct, it's quite interesting, but did you check systematically for the possibility of SNPs in the ECS and found it to be less frequent? Can one exclude the possibility of errors in ECS identification?
8. One of the important points of the paper is the comparison of genetic and environmental regulation. It is only natural to expect some quantitative comparison of the differences in editing between the strains at an equal T, vs. the differences between 18C and 25C in the same strain. Which makes a bigger change? One can look at a global editing measure, and compare the changes, or count the number of differentially edited sites taking into account sample sizes, etc.
9. Interestingly, the authors find that sites which show little change in editing between temperatures have significantly lower predicted free energies. What about sites that do change, but to the opposite direction (more editing in high T)?
10. Fig 3C: in the legend – change "Significant sites" to "Differentially edited sites". Please add the background level of the genetic difference (as found far away from editing sites). Did you consider (in the x-axis) the distance to a *nearest* editing site?

We thank you and the reviewers for the prompt processing and constructive feedback of our manuscript. Please see below for our point-by-point responses to reviewers' comments and suggestions. We have also made changes to our manuscript based on these comments and have highlighted the main changes in red for your convenience.

Reviewers' comments:

Reviewer #1 (Remarks to the Author):

Regulation of Gene Expression and RNA editing in *Drosophila* adapting to divergent microclimates at Evolution Canyon

Yablonovitch et al.

In this study the authors examine *Drosophila* from two microclimates within Evolution Canyon: the hot, dry South Facing Slope (SFS) and the cool, humid North Facing Slope (NFS). Whole-genome sequencing (WGS), RNA-seq, and microfluidic multiplex PCR-seq (mmPCR-seq) was performed on 16 isofemale lines from each microclimate. In the first set of experiments, the authors grew both populations of flies at 25°C and identified differences in genomic sequence, gene expression and RNA editing between the SFS and NFS flies. Approximately 500 transcripts were differentially expressed in both head and whole body samples. These gene expression differences could be partially explained by genetic differences between SFS and NFS flies: This is because there was an enrichment of SNPs in the promoters or coding sequence of differentially expressed genes compared to those genes with similar expression levels.

The authors then examined RNA editing in the two populations of flies and found ~60 differentially edited sites. Interestingly, the regions around these sites are also enriched for SNPs compared to sites that were not differentially edited. Previous studies indicate that editing levels can be modulated by mutations that modify the structure of the editing complementary sequence (ECS). Indeed, the authors convincingly show that mutations of the SNPs in ECS disrupt editing at two sites in one gene, *Prominin*. Finally the authors grow the two populations of flies under conditions that mimic the two microclimates, to test the impact of environment on RNA editing levels. They show that most of the editing differences observed between SFS and NFS flies at 25°C are not further exacerbated with temperature changes. (There is one exception to this statement: *falafel* shows temperature dependent RNA editing.) This suggests that the genetic variation between the two populations is responsible for the differences in editing, potentially differences in structure surrounding the differentially edited sites.

This study more generally addresses an interesting and outstanding question in the field of RNA editing: what is the mechanism behind the RNA editing differences observed in different populations of organisms (e.g. octopus populations @ different temps with differential editing)? The observation that the differential editing between SFS and NFS flies might be due to genetic changes within the ECS that occur during the divergent evolution of these two populations rather than a short-term or immediate environmental effect on editing is surprising and important. Unfortunately, the reader is distracted from this key result by the inclusion of less significant data. A shorter, more focused manuscript highlighting these editing findings would be stronger.

We acknowledge that the manuscript is mostly focused on the editing results. However, we think the gene expression results are still worth including, since we come to a similar conclusion as for the editing results (that is, that gene expression differences between the fly populations are associated with genetic differences in *cis*). We think that the gene expression results are also of interest to scientists from a broader range of fields within biology/genetics.

In addition, the authors need to address the major concern below prior to publication.

Major Concern:

One of the key conclusions of the paper is based on two observations: 1) that differentially edited sites have genomic variation in the regions surrounding the site and 2) these variations or SNPs in the genomic sequence modify the editing frequency by changing the secondary structure of the ECS. The key figure supporting observation #1 (Fig. 3c) is hard to interpret. It is not clear from the methods what editing sites were included in this analysis: 60 differentially edited sites from whole body and head or just the 23 sites showing a >5% change in editing frequency?

We apologize for the confusion. For this analysis, we originally compared the significantly differentially edited sites (51 sites from whole body mmPCR-seq + 9 sites from head RNA-seq = 60 all together) with the non-differentially editing sites (676 sites from whole body mmPCR-seq). However, we have since changed it so that the differentially edited sites only include significant sites which also have at least a 5% editing level difference between the NFS1 and SFS flies. This reduces the number of differentially edited sites used from 60 to 32 sites. We also considered that some of these sites are clustered close together, which could confound our results. Therefore, for clustered sites, we only included one representative site within each cluster in the plot. Specifically, if editing sites were clustered together (within 5kb of each other), then either the editing site with the greatest editing level difference between the NFS1 and SFS was included for the plot (for the differentially edited sites), or the editing site with the smallest editing level difference was chosen (for the non-differentially edited sites). Thus, in the end, we included 23 differentially edited sites and 283 non-differentially edited sites in the plot. In addition, there is still a significant enrichment of genetic differentiation near the differentially edited sites (p -value = 0.030). We have now added the details above to the Methods section (“RNA editing SNP enrichment analysis” section, page 19).

The y-axis of Fig. 3c shows tiny genetic differences of an aggregate plot. How many of the 60 differentially edited sites have one or more SNPs in the region surrounding a site, and how many have no SNPs. The authors have to present a more thorough and skeptical comparison between sites that are differentially edited and those that are not.

We thank the reviewer for this comment, and agree that more analyses would help strengthen our conclusion about the importance of genetic *cis*-regulation of editing between the Evolution Canyon fly populations. To study this further, we determined which sites have at least one highly differentiated SNP (top 0.5% F_{st} , similar to the gene expression analyses) within 100 bases, and within 500 bases of it, and compared this between the differentially edited and non-

differentially edited sites. We first performed a similar analysis as described above to remove clustered sites, but used a “clustering” distance of 100 bases or 500 bases, rather than 5kb. In addition, also similar to what we described above, the differentially edited sites only contain significant sites with at least >5% editing level difference between the NFS1 and SFS fly populations. When we consider SNPs within 100 bases of editing sites, we find that 3/26 (11.5%) of differentially edited sites have at least one highly differentiated SNP, whereas 2/378 (5.3%) of non-differentially edited sites have one. This is significant using a one-sided Fisher’s Exact Test (p-value = 0.0022). When we consider SNPs within 500 bases of editing sites, we find that 4/24 (16.7%) differentially edited sites have at least one highly differentiated SNP, whereas 14/332 (4.2%) of non-differentially edited sites have one. This is also significant using a one-sided Fisher’s Exact Test (p-value = 0.02574). Therefore, although we had limited success to identify differentially edited sites nearby highly differentiated SNPs, our analysis further suggests that SNPs nearby differentially edited sites are playing an important role in their editing level differences between the NFS1 and SFS flies. This analysis (using the distance of 100 bases) has been added to the manuscript on page 9, lines 236-240: “We specifically identified how many sites have at least one highly differentiated SNP (top 0.5% F_{st}) within 100 bases...”

The second key observation, i.e., that the genetic differentiation around the editing sites disrupted the ECS was only shown for the Prominin transcript. How many of the differentially edited sites have genetic changes that disrupt the predicted ECS? In other words, how many of the surrounding SNPs are in the ECS, i.e., how many of the 60 sites might be like the Prominin editing? Although the Prominin example is convincing, it is not clear that genetic variation in the ECS is the only or even the predominant mechanism driving differential editing.

We thank the reviewer for this comment, and also agree that it would be important to point out all of the differentially edited sites (q-value < 0.05 and more than 5% editing level difference) which have a highly differentiated SNP (top 0.5% F_{st}) in their respective ECSs. Similar to what was described above, we chose one representative site if sites were clustered within 100 bases of each other. Of the 26 differentially edited sites tested, we find that three differentially edited sites (5 if you include the clustered sites) have at least one SNP in their respective ECSs: the two *prominin* sites already described in the manuscript, two sites in the 3’UTR *falafel* (one of which was pointed out in the manuscript for showing a genotype-environment interaction), and one site in the 3’UTR of *Slob*. On the other hand, none of the 378 non-differentially edited sites had a SNP in their respective ECSs. Thus, the differentially edited sites show a significant enrichment of ECS SNPs (p-value = 0.00024, one-sided Fisher’s Exact Test). This analysis has been added to the manuscript on page 9, lines 240-243: “Similarly, we found that 3/26 (11.5%) of differentially edited sites have at least one highly differentiated SNP in their respective predicted ECSs...”. We note that our limited success to identify differentially edited sites with surrounding SNPs in the ECSs could be due primarily to two reasons. One reason is our limitation of predicting ECSs: for the 26 differentially edited sites test, we were only able to predict an ECS for 18 of those sites. The other reason is that the SNPs that affect editing may not always be located in the ECSs. This would explain the relatively small inter-slope differences of editing levels that we observed overall.

Minor concerns:

1) The authors state that the changes in editing levels at higher temperatures are “not due to decreases in Adar transcript levels at 25°C”. Previous studies indicate that Adar protein levels decrease with temperature with dramatic differences between 20°C and 30°C (Reider et al., 2015). Does a change in Adar protein levels explain the editing differences?

This would not be feasible for us to test in the Evolution Canyon flies, as there is no good antibody available for *Drosophila* Adar. Reider et al (2015) used an HA-tagged Adar fly line to measure Adar protein levels between different temperatures using the HA tag antibody. Another difference between our studies is that we tested the differences between temperatures 18°C and 25°C, while they used 10°C, 20°C and 30°C. It is possible that Adar protein level decreases at 30°C, but not at 25°C, compared to 18°C and 20°C.

2) On page 7 the authors state: “The RNA-seq data allowed us to identify several differentially edited sites that were not covered in the mm-Seq assay, including non-synonymous, intronic and intergenic sites”. This was confusing. Were these additional sites including in the analysis of differentially edited sites? Are they in addition to or included in the 51 sites from whole bodies and the 9 sites from heads?

We apologize for the confusion. Yes, these additional significant sites identified from the head RNA-seq data were included in the analysis of differentially edited sites (for instance, in Figure 3c). The sites referenced in the sentence are the 9 sites identified from the head RNA-seq data. The manuscript has been updated to clarify this on page 8, lines 219-223: “We also calculated differentially edited sites from the RNA-seq data of fly head tissue...”

3) How many editing sites show >5% difference when comparing SFS and NFS flies. Is it the 23 genes shown in Fig. 3a? Are these the same genes as in Fig. 5D (but there are 21 genes here). The figure legend of Fig. 5D says these are the same genes as Fig 2c?

The number of editing sites in Figure 5d is decreased by two because those two sites did not have adequate coverage in enough samples in the 18°C mmPCR data (at least 50 reads for at least 3 samples for both the NFS1 and SFS groups). The Figure legend for 5d has been updated to reflect this.

Reviewer #2 (Remarks to the Author):

The topic of this study is trendy and interesting. Yablonovitch et al. want to detect SNPs with divergent allele trajectories between North and South populations (measured as F_{st}) and study whether candidate SNPs can explain divergent RNA editing and differential expression levels.

I have several important concerns about this study that would make me prevent this manuscript from publication.

Pp5, 114-126: The removal of NSF2 from the analysis is not justified. PC2 clearly separates NSF2 from SFS. The analysis should include NSF2 or at least two analyses should be done: one

including NSF2 and other excluding it.

We thank the reviewer for this comment. As we stated in the manuscript, we focused on the NFS1 and SFS fly populations because PC1 (which explains the most variation in the data set) separates out NFS1 and SFS, and because there are more flies in NFS1 and NFS2 (thus suggesting that the NFS1 flies may represent a more accurate representation of the flies living on that slope). Nevertheless, we have determined the numbers of differentially expressed genes and differentially edited sites with NFS2 included, and have included this in the supplementary data (Supplementary Data 1), and have also mentioned it in the manuscript on page 5, lines 120-123: “We have included a list of differentially expressed genes (from whole body and head RNA-seq) and differentially edited sites (from 25°C whole body mmPCR-seq) between the combined set of NFS1 and NFS2 flies, compared to the SFS flies”. For whole body gene expression, we observed the number of significantly differentially expressed genes decrease from 515 genes to 213 genes; for head gene expression, the significant genes went down from 449 to 317. For 25°C whole body mmPCR-seq RNA editing, we observed the number of significantly differentially edited sites between the fly populations to decrease from 51 to 12; for RNA-seq editing (which was from fly heads), the number of significant sites decreased from 9 to 0. Since the flies in the NFS1 and NFS2 are distinct from each other, this decreases our power to identify differences in gene expression and editing. As a result, all remaining analyses in the paper that compare the two groups will compare NFS1 and SFS.

Pp5, 127-134: the authors mention a 9.71Mb sweep region, 5.85Mb of it as soft sweep region. Where is this region located? Why the author only mention this one when there are many more in suppl. Data 1? Where can I see this region in figure 1c? What is the point of all this paragraph?

The sweep region is not one single, continuous region in the genome; it is simply the sum of all of the sweep regions that were identified (included in the new Supplementary Data 2). This has been clarified in the manuscript on page 6, lines 140-144: “Sweep regions with total length 9.71 Mb were found in the NFS1 population...”. We have added a new supplementary figure (Supp. Figure 1) that visualizes where the sweeps are located along each chromosome. The purpose of the paragraph is to confirm that the NFS1 and SFS groups are experiencing evolutionary changes, and we’ve clarified this in the manuscript on page 6, lines 144-146: “A visualization of where these sweeps are located along each chromosome is presented in Supplementary Figure 1, and demonstrates that the NFS1 and SFS flies have experienced unique evolutionary changes.”

Pp6, 148-151: “In addition, we examined genes involved in pigmentation, DNA repair and heat stress response, as one might expect these genes to be more highly expressed in flies lining in the SFS...”. Why? These GO terms or related GO terms do not seem to have an enrichment of DE genes. Is this looks like an ad hoc analysis. If so, please justify why. In addition: What genes are those? How did the authors select them? According to a GO terms? If so, which GO terms? Please provide numbers.

These are genes that have previously been shown to play a role in adaptation to different environments, both within and outside Evolution Canyon; these citations have been added to the manuscript (page 6, line 164). Even though these gene categories did not show GO term enrichment, we were still interested in determining how many genes in these categories showed

differential expression in our data set, given their previously-established roles in adaptation. We chose the genes based on the GO terms of the gene categories we mentioned (pigmentation, DNA repair, heat response). We've added the GO term numbers we used in the Methods section ("Differential gene expression analyses" section, page 17-18).

Pp6, 156-164: I don't see any clustering in figure 2e. There is no way to split both populations in this figure.

In Figure 2e, the populations are separated along the PC2 axis. Although the clusters are not as distinct as in Figure 2f, there is still a clustering pattern in the plot.

In addition, the authors say in pp5 – 123: "use only these 8 NFS lines in subsequent analyses", which is certainly not the case in figure 2c and others. Am I missing something?

We apologize for the confusion. We have updated this sentence to say that we focused on these fly lines for subsequent analyses (page 5, lines 126-127: "Thus, we focused on these 8 NFS lines to represent the NFS fly population in subsequent analyses.") We also clarified before discussing the gene expression and RNA editing PCA results that we used all Evolution Canyon lines for the PCAs.

In addition, I don't understand the rationale in using PCA to prove that DE is due to genetic differentiation. If the two groups show DE at the same conditions (temperature) is because there is a genetic component. Why then do the authors need to do the PCA?

While we agree that a genetic component is responsible for gene expression differences between the Evolution Canyon fly populations, the PCA plots go further and examine whether the gene expression patterns cluster the fly samples in a similar manner as the SNP patterns. As we stated in the main text (page 7, lines 172-174): "A previous study which used PCA to compare genetic and gene expression differentiation between different human populations did not show similar groupings between the two data sets". Specifically, this study identified clustering by population in the SNP PCA, but observed no clustering in the gene expression PCA. So, even though the samples are grown at the same conditions, this previous study demonstrates that the patterning in a SNP PCA can indeed be different from the patterning in the corresponding gene expression PCA, suggesting there is not always such a clear association between genetic and gene expression differences.

In addition, what data is used for the PCA, DE genes or all expressed genes?

All expressed genes are included in the PCA. The Methods section has been updated to reflect this ("Principal Component Analysis (PCA)", page 17).

Pp6, 167: the authors should provide some summary statistics for the Fst. For instance, what is the rank of the top 0.5% Fst. In addition, I guess $\sim 2,000,000 \times 0.005 = 10,000$ is the number of candidate SNPs. How do the authors handle this data? In addition, what are these numbers: 77/428, etc. I guess 77 are DE genes that have at least 1 candidate SNP and 428 are DE with no candidate SNPs? If so, the authors must consider a bias due to gene length effect: i.e., the longer

the gene the more likely it will have a candidate SNP just by chance. The authors may want to use similar approaches as those used to study enrichment of SNPs in GO terms.

We thank the reviewer for these comments. If by “rank” the reviewer means the top 0.5% Fst threshold, this corresponds to $F_{st} = 0.766$. Yes, there are approximately 10,000 candidate SNPs. In terms of how we handle the data, we used this data for SNP enrichment analyses for gene expression and RNA editing (see “SNP enrichment in differentially expressed genes” and “RNA editing SNP enrichment analysis” Methods sections for details). Yes, the reviewer is correct in their interpretation of the gene expression numbers. We also agree that gene length could bias the SNP enrichment analysis for differentially expected genes; thus, we changed our analysis method and used GSeq (which considers gene length) to determine whether differentially expressed genes show a significant enrichment of SNPs. Overall, we still see a significant SNP enrichment for both head and whole body differentially expressed genes: p-value = 0.00026 for head, and p-value = $1.26e-06$ for whole body. The manuscript has been updated accordingly on page 7, lines 182-186: “Using the R package GSeq²², which takes into account gene length...”.

Pp6, 171: “this provides evidence that genetic mutations within the gene of interest are contributing to gene expression differences...”. So, if I properly understand this, this result implies that SNPs occurring at genes are likely involved in regulation of gene expression? To my understanding from other studies, most gene expression divergence within populations and between species is do to mutations/SNPs occurring in regulatory regions, not in the gene body.

We agree that gene expression divergence is more likely due to SNPs in regulatory/non-coding regions of genes, rather than the gene body itself. However, SNPs in coding regions may still affect gene expression by altering RNA processing and stability, translation initiation and elongation, and protein folding. In addition, we have included an additional analysis in the manuscript to determine whether there is SNP enrichment in the 1kb region upstream from genes (potential promoter region) on page 7, lines 186-193: “In addition, when we examine potential promoter regions of genes (1kb upstream), we observe a similar enrichment...”. For significantly differentially expressed head genes, we find 30/424 (7.1%) of these genes have at least one highly differentiated SNP (top 0.5% Fst) in the upstream 1kb region, whereas 491/10900 (4.5%) of non-significant genes do. This is a significant enrichment (p-value = 0.013, one-sided Fisher’s Exact Test). For significantly differentially expressed whole body genes, we find 44/493 (8.9%) of these genes have at least one SNP in the upstream 1kb region, whereas 637/13394 (4.8%) of non-significant genes do. This is also a significant enrichment (p-value = $8.8e-05$, one-sided Fisher’s Exact Test). Therefore, both SNPs in the gene body, as well in putative promoter regions, are enriched in the significantly differentially expressed genes.

Pp7, first paragraph: finding candidate SNPs in upstream regions of genes located at selected regions (sweeps) does not prove o indicates these are adaptive regulatory mutations. For instance: how many DE genes in these sweep blocks do not have candidate SNPs. How many candidate SNPs are in -1kb of equally expressed genes located in these sweep blocks?

We observed 63 whole body differentially expressed genes in sweep blocks that do not have these candidate SNPs/indels, and 68 head differentially expressed genes in sweep blocks that do not have these candidate SNPs/indels. Equally expressed whole body genes in sweep blocks

have 165 candidate SNPs/indels -1kb of their transcription start sites, and equally expressed head genes in sweep blocks have 126 candidate SNPs/indels -1kb of their transcription start sites. However, we are not trying to claim that the mutations we highlighted are definitely adaptive, only that they are possible candidates. We have changed the wording slightly in the first sentence of this paragraph to clarify this on page 7, lines 196-197: “We next wanted to determine which genes were most likely to have potentially adaptive regulatory mutations responsible for gene expression differences between the NFS1 and SFS flies”. In addition, the last sentence of this paragraph states that these are just candidates for adaptation, and future studies will need to be done for validation (page 8, lines 204-205: “These genes and their associated mutations are candidates for future validation experiments of gene expression regulation and adaptation in Evolution Canyon flies.”)

Pp7, last paragraph: This concern about PCA is similar as in Pp6, 156-164. It looks like there is some differentiation between NFS1 and south population in PC1, but not in PC2.

Although there is only differentiation along the PC1 axis, we think this result is still worth sharing. It further supports the claim that genetic differences are associated with RNA editing differences between the NFS1 and SFS fly populations.

Pp8, first paragraph: it is difficult to evaluate figure 3c. 1) why do the authors need to normalize Fst? I think the authors should use the actual Fst.

The Fst needs to be normalized for the number of sites in each group (significantly differentially edited sites, and non-differentially edited sites). We changed the word to “average” in the main text of the manuscript to make this clearer.

2) higher Fst around significant sites does not prove a cis-effect in editing.

We agree that higher Fst values around significant sites does not prove that *cis*-regulation is playing a role in the editing level differences for all of these sites. However, we think it is reasonable to claim that a significant enrichment of genetic differentiation around the significant sites, compared to the non-significant sites, suggests that these genetic differences are likely to be playing a role in regulating these editing level differences.

3) these lines represent averages. The authors should also plot the CI at 95%. KS test does not consider this variance.

Please see Figure R1 below, which has the 95% CI added to it. (Note: we also changed what we considered “differentially edited” sites in this plot. While previously it was all sites which had q-value < 0.05, we have updated it to include all sites which have q-value < 0.05 and >5% editing level difference between NFS1 and SFS. There is still a significant enrichment of genetic differentiation near the differentially edited sites (p-value = 0.030)).

Figure R1. Adapted from Figure 3c in manuscript. Plot showing enrichment of differentiated SNPs near significantly differentially edited sites (p -value = 0.030, one-sided KS test), along with 95% confidence intervals (C.I.).

The 95% confidence interval for the non-differentially edited sites is relatively narrow compared to that for the differentially edited sites. However, this makes sense as there are more than 10 times fewer differentially edited sites (23 differentially edited sites vs. 283 non-differentially edited). In addition, we expect a certain amount of variation because the SNPs which affect editing are not always the same distance away from the editing sites. Thus, because of this, and the additional analyses we added to the manuscript (see Reviewer #1 comments), we believe that it is still valid to claim that there is a significant enrichment of genetic differentiation around differentially edited sites.

Pp9, 258-260: this proves that ECS is involved in editing efficiency, but it does not prove that the alternative aimed SNP is adaptive.

Sentence referenced: “Overall, this demonstrates that the *prominin* ECS is a key structural element needed to maintain high *prominin* editing levels, and that the SNP we identified in the ECS helps account for the difference in *prominin* editing levels between the NFS1 and SFS flies.” We agree that it does not prove that the SNP is adaptive, and we do not claim this in the manuscript.

Pp9-11, section “Comparing genetic and environmental regulation of editing in evolution...”. If the authors want to study the genetic and environmental (temperature) components of RNA-editing variation they should fit the data to a linear model (LM, if the data is normally distributed) or to a generalized linear model (GLM, which is likely the correct one given the binary nature of the data – edited read v unedited read). A GLM will allow the authors to break the RNA editing

differentiation into the different factors: $G + T + G \times T$. I don't think there is any other alternative to this.

We thank the reviewer for this comment, and apologize that our methods for this analysis were not clear enough. We agree that the data should be fit to a linear model or generalized linear model. For our analysis to identify genotype-environment interactions of editing, we originally fit the data to a linear model, followed by an ANOVA test. As stated in the manuscript, we observe one site (in *falafel*) which shows a significant interaction at q -value < 0.1 . When we fit to a generalized linear model instead, we get the same results, and the same q -value (0.065). We have updated the manuscript accordingly (see new "Genotype-Environment Interaction Analysis" Methods section, page 20).

Pp15, 438: why do you filter sites with $< 20\%$ difference between replicates? Idem for the $> 98\%$ and $< 2\%$. You might be biasing your analysis.

As stated in the methods, we included sites with $< 20\%$ difference between biological replicates. The sites with $\geq 20\%$ editing between biological replicates do not have reliable, consistent measurements, and thus were not included in the analysis. Based on repeated experiments in our lab with mmPCR-seq, we are fairly confident that these large changes are the result of technical variation rather than biological variation. For the $> 98\%$ filter, we removed this as it actually did not filter out any additional sites, and updated the Methods accordingly. We filtered out sites that were lowly edited because we initially examined editing at all sites that have ever been known to be edited. However, some of these sites have very low editing ($< 2\%$) across all samples in our analysis, so we excluded them as candidates for differential editing. A similar type of filtering is also done for gene expression comparisons, which this reviewer mentions three comments down.

Pp15, 443: what is identical-by-descent and how do you compute it?

Identical-by-descent is a region of DNA that is shared between two or more individuals and is inherited from a common ancestor. Thus, if there is high IBD between individuals, they share some relatedness. IBD was computed using PLINK's identity-by-descent function. The manuscript has been updated accordingly ("Identification of putative selective sweeps" in Methods section, page 17).

Pp15, 451-457: what is a hard sweep and what is a soft sweep.

In a hard sweep, one newly occurring mutation is rapidly swept up to fixation in a population. In a soft sweep, multiple different mutations are simultaneously swept up in a population. This has been incorporated into the manuscript on pages 5-6, lines 138-140: "In a hard sweep, one newly occurring mutation is rapidly swept up to fixation in a population, while in a soft sweep, multiple independent mutations are simultaneously swept up in a population".

Pp16, Section "SNP enrichment in differentially expressed genes": "... we first filter out lowly-expressed genes from non-significant genes". Why? Do you also filter out lowly-expressed genes from significant genes? I'd strongly suggest doing such a filter before the DE analysis so

that lowly-expressed or unexpressed genes are not included in the analysis (see edgeR manual). This is important as the FDR value depends on the number of tests.

We used DESeq2 to identify differentially expressed genes. DESeq2 uses an algorithm to automatically filter out lowly expressed genes. If we filter out genes with ≤ 10 read counts across all samples prior to the DE analysis, as the EdgeR manual recommends, we have the same number of differentially expressed genes for NFS1 vs. SFS whole body, and two fewer differentially expressed genes for NFS1 vs. SFS heads. Since this changes the number of differentially expressed genes by $< 0.5\%$, we are keeping the original set of differentially expressed genes we identified.

“we only included genes which had high enough expression to generate a p-value...” What is high enough?

As stated above, this criterion is set by DESeq2’s algorithm. After manually observing the average read count for those without a p-value, they all had < 1 average read count among the samples. We have explained this in the text in the “SNP enrichment in differentially expressed genes” section on page 18: “specifically, we only included genes which had high enough expression to generate a p-value of differential expression from DESeq2, which is determined by an algorithm in the DESeq2 software.”

“We excluded genes non-significant genes whose length was shorter than the minimum significant gene length or longer than the maximum significant gene length” What is minimum and maximum significant gene lengths?

We changed our analysis, and used GSeq instead, which takes gene length into account (see details in response to previous comment).

Pp16, 478: again, the authors are manipulating the data with any reason to do this. This might bias the analysis significantly and the conclusions.

We addressed why we filtered out sites with more than 20% difference in editing between biological replicates in response to a previous comment of this reviewer. In regards to why we used a coverage cut-off of 50, this is necessary as there technical variation in the mmPCR-seq editing levels, which becomes more noticeable at lower coverages. There is also a balance between comparing too few sites (we might be missing some interesting sites if we require all samples to have a certain amount of coverage), versus too many sites (sites with very few samples represented and/or low coverage are unlikely to be significant). We have also used similar cut-offs in some of our previous publications (for example, see Ramaswami, Deng, *et al*, *Nature Communications*, 2015).

Pp16, 480-483: the authors use a t-test. Here, and in the whole manuscript, the authors must justify that the t-test is the appropriate statistical framework. For instance, is editing level normally distributed? In addition, the authors apply Storey and Tibshirani’s adjustment method for multiple testing (qvalue) and afterwards apply BH correction. This makes no sense.

We looked into whether the editing levels for each site tested are normally distributed. Using the Shapiro-Wilk test, we see that 143/727 sites (SFS 25°C editing levels), 56/727 sites (NFS1 25°C editing levels), 144/709 sites (SFS 18°C editing levels), and 61/677 sites (NFS1 18°C editing levels) come from a non-normal distribution. Thus, approximately 7-20% of sites come from a non-normal distribution. However, for the following two reasons, we think our use of the t-test in the manuscript is justified:

1) We used the Mann-Whitney-Wilcoxon test, a nonparametric test which does not have the normality assumption, to identify significantly differentially edited sites between the NFS1 and SFS fly populations, and compared with the results from the t-test. We observed 25 significantly differentially edited sites (q -value < 0.05), including the sites in *prominin* and *falafel* that we highlighted in our manuscript. Since the Mann-Whitney-Wilcoxon test is not as powerful as the t-test, the lower number of significant sites is somewhat to be expected. When we increase the q -value threshold to 0.1, we observe 52 significantly differentially edited sites. We found that 45 of these sites overlapped with the 51 significant sites we identified using the t-test with q -value < 0.05 (88%). Thus, both tests yield similar results.

2) For almost all of the plots and analyses we performed that utilized the significant editing sites, we included an additional threshold of at least 5% editing level difference between the different fly populations, to take into account that some of our significant sites are likely to be false positives. These include Figure 3a, 5b, 5c, and 5d. The major analysis where we originally included all significant sites, regardless of editing level difference, was in Figure 3c (showing higher genetic differentiation around significant sites). However, as stated earlier, we have since updated the plot in the manuscript so that only significant sites with at least 5% editing level difference between NFS1 and SFS are included (see Figure R2 below).

Figure R2. Plot showing enrichment of differentiated SNPs near significantly differentially edited sites (p -value = 0.030, one-sided KS test). Significant sites included have at least 5% editing level difference between NFS1 and SFS flies.

We still observe significantly higher genetic differentiation around these significant sites with >5% editing level difference between the NFS1 and SFS flies (p-value = 0.030, one-sided KS test). Thus, even though we found the editing levels of some sites to be non-normal, we ensure that only the significant and most differentially edited sites are included in subsequent analyses.

In regards to other assumptions of the t-test, each of the samples is a random sample of its population (from the North Facing Slope or South Facing Slope), and equal variances were not assumed in the test (Welch's t-test).

We apologize for the confusion with the multiple testing correction methods. We only considered sites with a q-value < 0.05, and did not apply the Benjamini-Hochberg correction after this. The manuscript has been updated accordingly (see "RNA editing comparison using mmPCR-seq data" in Methods section, page 18).

Pp16, 487: "we tested for significant differences in editing using a likelihood ratio test". How are the two likelihoods needed for the LRT computed?

We use a Beta-Binomial model to test the differential editing level. At each site, the underlying editing level, π , in each strain is modeled as a Beta distribution. Conventionally Beta(a, b) probability density function is written as

$$f(\pi; a, b) = \frac{\pi^{a-1}(1-\pi)^{b-1}}{B(a, b)}$$

where

$$B(a, b) = \frac{\Gamma(a)\Gamma(b)}{\Gamma(a+b)}.$$

We re-parametrize this density function by Beta(u, N), such that $u = \frac{a}{a+b}$ and $N = a + b$. For strain i , conditioning on the total reads mapped to the site, X_i , the number of the edited reads, X'_i , is binomial: $X'_i | X_i \sim \text{Binom}(X_i, \pi_i)$. Marginally X'_i has a Beta-Binomial distribution.

Under the null hypothesis of equal editing level, $\pi_i \sim \text{Beta}(u_0, N)$ for all strains; in other words, all π_i are drawn from the same Beta distribution. Under the alternative hypothesis, we let $\pi_i \sim \text{Beta}(u_1, N)$ for SFS flies and $\pi_i \sim \text{Beta}(u_2, N)$ for NFS flies. The parameters, (u_0, N) under the null hypothesis and (u_1, u_2, N) under the alternative hypothesis are estimated by maximizing the joint likelihood given all observed RNA-seq editing data. We note that this model accounts for over-dispersion in the count data by allowing π_i to vary between strains.

Pp17, 497-500: "To test whether there was higher SNP enrichment for the significant sites, we averaged this normalized Fst value over 250bp non-overlapping windows and performed a one-sided KS-test". This DOES NOT prove that you have more SNPs, only proves that you have different Fst for the SNPs.

We changed this to: "To test whether there was higher genetic differentiation around the differentially edited sites..." ("RNA editing SNP enrichment analysis" Methods section, page 19).

MINOR CONCERNS

Pp5, 127: the authors should explain what is a selective sweep, what is results is expected (reduction of nucleotide diversity, etc), and what is the difference between hard and soft sweeps.

The manuscript has been updated to reflect this on page 5, lines 131-133: “Selective sweeps are regions of DNA that show reduced nucleotide diversity, as a result of one or more beneficial alleles in that region reaching fixation in the population.”, and pages 5-6, lines 138-140: “In a hard sweep, one newly occurring mutation is rapidly swept up to fixation in a population, while in a soft sweep, multiple independent mutations are simultaneously swept up in a population”.

Pp7, 157: what is the RNA-Seq data set?

We’re not sure exactly what the reviewer is asking here. We used gene expression measurements from both head and whole body RNA-seq data to generate the PCA plots, as described in the text here.

Pp7, 175: “after overlapping these genes...” How many gene are “these genes”? How many do you expect to find?

“These genes” refers to significantly differentially expressed genes that have at least one highly differentiated SNP or indel in the 5’UTR, 3’UTR, and in the region 1kb upstream of the transcription start site. However, we have changed some of the wording in this paragraph to introduce a dataset recommended by Reviewer #4 (see page 7, lines 198-199: “To do this, we first identified the highly differentiated (top 0.5% F_{st}) SNPs and Indels that overlapped with a selective sweep in the NFS1 or SFS flies (Supplementary Data 4)...”), so we are no longer referring to these specific genes in this paragraph. Still, to answer the reviewer’s questions, if we examine head differentially expressed genes, we observe 71 genes in this category. If we examine whole body differentially expressed genes, we observe 82 genes in this category. The number we expect to find is the proportion of all genes that have at least one differentiated SNP in these gene regions, multiplied by the number of significant genes. This comes out to 45 head genes, and 51 whole body genes. Therefore, we find more differentially genes than expected have at least one differentiated SNP in these gene regions, although it is possible that the different lengths of these gene regions (between the significant and non-significant genes) may bias this analysis slightly.

Pp7, 196: how much is “several”?

7 of the 9 significant sites identified from the head RNA-seq data were not covered by the mmPCR-seq assay. This has been added to the manuscript on page 8, lines 221-223: “In total, we identified 9 differentially edited sites from the RNA-seq data of fly head tissue...”

Pp8, 230-231: I don’t understand this sentence.

The original sentence is: “Intriguingly, we found one differentiated SNP in the *prominin* ECS, which occurred in 7 out of the 8 NFS1 lines, and in none of the remaining lines.” Although we are not sure exactly what the reviewer doesn’t understand about this sentence, we have changed

it to: “Intriguingly, we found one differentiated SNP in the *prominin* ECS, which was in 7 out of the 8 NFS1 lines, and in none of the remaining Evolution Canyon lines.” (page 10, lines 272-274).

Pp8, 234 and on: MUTANT allele, instead of mutated allele? In addition, can you maybe provide a name for the mutant and the alleles. Otherwise is very confusing.

“Mutated allele” has been changed to “mutant allele”. We also updated the manuscript with the following mutant names: “*prom* ECS deletion mutant”, “*prom* ECS SNP mutant”, “*prom* PAM mutant”.

Pp15, 434: what is the genome coverage. This will help to evaluated the minimum cut of 5 reads.

Please see Figure R3 below for a histogram of the average genome coverage (averaged across all Evolution Canyon samples).

Figure R3. Histogram showing average genome coverage distribution for the 32 Evolution Canyon lines.

Reviewer #3 (Remarks to the Author):

The ability to recode has been one of the most extensively studied aspects of RNA editing. Most of the seminal studies focused on how individual recoding events affected protein function. A basic question regarding recoding is why would an organism choose to alter RNA instead of DNA, the genomic blueprint? The obvious, and generally assumed answer is that RNA-level changes provide flexibility: RNA edits can be turned on or off, generating distinct protein isoforms. It has been hypothesized that this plasticity might be in response to environmental conditions, but few studies have addressed this directly. Yablonovitch et al add important data to this theme. Past studies have shown that RNA editing correlates with environmental variation. Using transcriptome wide screens of editing sites, this paper looks at plastic sites between

Drosophila populations and asks which are due to acclimation and which are due to adaptation. Further, it asks how the adaptive sights might be generated. I think that the paper makes important contributions. However, there are some points that need to be addressed to strengthen conclusions and areas where the paper could be presented more clearly. The paper should be interesting to a broad audience.

Statistical analysis is appropriate

-Joshua Rosenthal, MBL

Major points:

1. At several places, the authors say that environmental differences act on editing in a more “global” manner. I’m not sure exactly what this means. The data in Fig 5b shows that while more sites are edited at a higher level at 18C, there are still plenty of examples of sites that are more highly edited at 25C. This is an important point because it’s a central theme of the paper. Maybe I’m missing something, but I just don’t see the changes as being global.

We apologize for the confusion. We originally used the term “global” to imply that more editing sites are affected by temperature than are affected by population genetic differences. We have removed this term and have changed the wording accordingly (for example, on page 11, lines 316-318: “However, we see more frequent, as well as larger changes in editing when we compare editing levels between flies raised at 18°C and 25°C, for both the NFS1 and SFS flies”)

In addition, it would be good to see how much overlap there is between the sites that significantly change with temperature between the NSF1 and SFS flies.

There are 173 sites that are differentially edited between temperatures among the NFS1 flies, and 166 sites that are differentially edited between temperatures among the SFS flies. The overlap between these is 137 sites. We have added this information to the manuscript, on page 11, lines 318-320: “Of the 173 differentially edited sites (q-value < 0.05 and >5% editing level difference) between 18°C and 25°C in the NFS1 flies, and the 166 differentially edited sites between 18°C and 25°C in the SFS flies, we find 137 sites that overlap.”

2. It would be good to have two more comparisons: how does this data compare with that presented in Reider et al.? Are the same sites changing with temperature? Also, at 25C, how well do the RNA editing levels (and sites) match between the evolution canyon strains and past data published for a lab strain (like Canton S)? It would be interesting to see how much plasticity there is in the editome and the authors have a good opportunity to show this.

We thank the reviewer for this great suggestion. Please see Figure R4 below for comparisons between editing level differences at different temperatures. The x-axis of each plot shows 18°C editing levels – 25°C editing levels for the SFS or NFS1 flies. The y-axis of each plot shows the corresponding data from Rieder *et al*, *BMC Biology*, 2015: either 20°C – 30°C editing levels, or 10°C – 20°C editing levels. There seems to be better correlation with our data and the editing level differences between 20°C and 30°C, than for the editing level differences between 10°C and 20°C. There is also a larger “spread” among editing level differences between their data and our data, which makes sense as they used a larger temperature difference.

Figure R4. Comparing temperature-dependent editing level differences between those demonstrated in this manuscript, and those demonstrated in Rieder *et al.* Specifically, (a) compares 18°C – 25°C editing level differences in the SFS flies, vs. 20°C – 30°C editing level differences in Canton-S flies, (b) compares 18°C – 25°C editing level differences in SFS flies, vs. 10°C – 20°C editing level differences in Canton-S flies, (c) compares 18°C – 25°C editing level differences in NFS1 flies, vs. 20°C – 30°C editing level differences in Canton-S, and (d) compares 18°C – 25°C editing level differences in NFS1 flies, vs. 10°C – 20°C editing level differences in Canton-S.

Please see Figure R5 below for editing level comparisons between a lab strain (UCSD stock: 14021-0231.69), whose editing levels were also measured using mmPCR-seq, and the average NFS1 and SFS editing levels. Most sites show similar editing levels between these groups. However, there are dozens of sites that show large editing differences between the groups, suggesting that the editing levels of certain sites can be quite plastic.

Figure R5. Comparing editing levels of a lab strain with editing levels from the NFS1 or SFS fly lines.

While we find these results interesting, we believe it is outside the scope of our manuscript and feel that it could be a distraction to readers. Therefore, we are not including the analysis.

3. Regarding the structure driving RNA editing levels of prominin sites 770 and 773, the authors state “that the SNP we identified in the ECS helps account for the difference in prominin editing levels between the NFS1 and SFS flies.” As the authors are aware, the additional mutation that was engineered to abolish the unwanted PAM site complicates matters because this mutation alone leads to a drastic reduction in editing (81%>21% at 770 and 79%>8% at 773). By adding the SNP on top of this editing is abolished. The authors conclude that this further reduction in editing by adding the SNP shows that it can reduce editing on its own. It is also possible that it only can reduce editing when in combination with the PAM mutation. It would be good if there were some independent verification that the SNP alone can reduce editing

We agree that it would be good to see that the SNP alone can reduce editing. We originally tried to make two independent CRISPR mutants to mutate the SNP in the Prominin ECS, with each having different guide RNAs and associated PAM mutations. However, after screening through the flies, we were only able to validate the SNP mutation in the ECS for one of these mutants. This mutant had a PAM mutation 10 bases away from the SNP, in the ECS, and is the mutant we presented in this manuscript. Unfortunately, the other independent CRISPR line to generate the ECS SNP mutant did not work. It would have had a PAM mutation in a different location 30 bases away from the SNP. Since this PAM mutation is further away from the SNP and the editing sites in the predicted RNA structure, it would have likely not affected editing levels as much, and we would have included the Prom editing levels for this mutant if it had worked. We have stated this caveat in the manuscript on page 14, lines 398-400: “...we note that we could not determine the exact amount of editing level regulation contributed by the SNP, since a CRISPR-associated PAM mutation was made in the same mutant.”

Minor points:

1. First sentence of abstract should be modified from “Determining the mechanisms by which a

species adapts to its unique environment is a key endeavor in evolution, ecology, and conservation” to “Determining the mechanisms by which a species adapts to its unique environment is a key endeavor in THE STUDY of evolution, ecology, and conservation.”

The manuscript has been updated accordingly.

2. “Inosine gets recognized as guanosine during translation, so A-to-I RNA editing has the potential to alter amino acid sequences of proteins and affect other transcriptional processes⁶.” A review shouldn’t be referenced here. The authors should reference the primary articles that support these themes.

Primary article references have replaced the review that was referenced (page 3, line 60).

3. The discussion of selective sweeps needs to be clarified and expanded. As it stands, it comes out of the blue. Many the readers of this paper will not be evolutionary biologists. For example, people from different areas of the RNA editing field will be interested in the findings and they may not be well versed in the concept of a selective sweep. A little more background and context is necessary.

We have added some background and context on selective sweeps in the manuscript on page 5, lines 130-133: “Next, we wanted to determine which regions of the genome are under selection, and thus likely to be adaptive...” and page 5-6, lines 138-140: “In a hard sweep, one newly occurring mutation is rapidly swept up to fixation in a population, while in a soft sweep, multiple independent mutations are simultaneously swept up in a population.”

4. Rob Reenan’s paper about the guided evolution of the *Drosophila* synaptotagmin site across insects (Nature. 2005;434 :409-13) should be added to the discussion at several places.

We have included this citation in the discussion (see page 14, line 406 and page 14, line 415).

5. For p5-6 when talking about GO term enrichment, make it clearer whether you’re talking about enrichment in NFS or SFS.

We have added clarification that the enrichment is for genes over-expressed in the SFS flies (page 6, lines 153-154).

6. When introducing the data on the “CRISPR mutants” a little general context is required. First of all, the name “CRISPR mutant” should be changed to something else. In addition, a couple sentences should be added explaining how the technique works and what it will do in this instance.

We updated the manuscript with the following mutant names: “*prom* ECS deletion mutant”, “*prom* ECS SNP mutant”, “*prom* PAM mutant”. We also added a couple of sentences about how CRISPR works and what it will do in this instance on page 10, lines 286-289: “By binding to a pre-designed guide RNA, the CRISPR-associated protein 9 (Cas9) endonuclease is directed to a specific region in the genome, where it can make precise genetic mutations³⁵. We used the

CRISPR-Cas9 system to make point mutations in the ECS, as well as remove the entire ECS sequence.” We also added another sentence in the “Generating *prominin* CRISPR mutants” Methods section (pages 19-20) to provide more detail.

7. A couple sentences should be added to more clearly explain what “falafel” does.

We have updated the manuscript to include more information about falafel on page 12, lines 349-351: “*Falafel* is a regulatory protein that plays a role in the cell cycle, specifically in the asymmetrical division of neuroblast cells³⁷. It has also been shown to be associated with DNA repair³⁸”.

8. Figure 5d is unclear. I understand that you are looking at sites that showed a difference between the SFS and NFS1 populations in figure 2. Then you acutely changed the temperature between 18C and 25C and looked for changes. But is this data from NFS1 flies or SFS flies?

The y-axis of this plot represents the SFS – NFS1 editing levels, at 25°C and 18°C separately. The manuscript has been updated to clarify this on page 12, lines 340-342: “For sites we identified as significantly differentially edited and have at least a 5% difference in editing between NFS1 and SFS, we plotted the change in editing between the NFS1 and SFS flies at 25°C and at 18°C.”

8. It should be pointed out that the quantification of editing using RNAseq data cannot accurately deconvolute editing from splicing. That’s to say, if different splice variants are edited at specific frequencies and their relative expression changes, this looks very much like changes in editing frequency within a splice variant. This should be mentioned as a caveat.

This caveat has been added to the manuscript on page 8, lines 223-226: “We note that the differentially edited sites identified from the RNA-seq data may be caused by a change in the relative expression of different splice variants that are edited at specific frequencies, rather than a change in editing within a particular splice variant.”

9. “Evolution Canyon is uniquely suited for studying biodiversity, evolution and 318 adaptation,”...this is probably an overstatement.

We have updated this part of the sentence to: “Evolution Canyon is uniquely suited for studying the biodiversity, evolution and adaptation of organisms that live in relatively close proximity to each other....” (page 13, lines 368-370).

Reviewer #4 (Remarks to the Author):

RNA editing contributes to the diversity of the drosophila proteome, to an extent far greater than in mammals, as was recently demonstrated by a number of groups. This post-transcriptional mechanism is well-suited for adaptation, as was demonstrated by Garret and Rosenthal (Ref. 10). However, the extent of adaptation through editing, and the mechanisms through which it occurs (genetic mutations vs. regulation) are still poorly understood. The present manuscript by

Yablonovitch et al is therefore an important and timely contribution.

Overall, the work is comprehensive, insightful, and well-written, and I find it suitable for publication in Nature Communications. However, I have a few questions, comments and suggestions that the author might want to consider prior to publication.

1. In p.5 (and support data 1) sweep regions are reported. Can one predict which newly fixated mutations can be associated with these regions, or at least report the mutations in these regions that distinct NFS and SFS lines?

We have included a new supplementary data file (Supp. Data 4) that contains the top 0.5% F_{st} mutations (SNPs and indels) that are in the sweep regions (NFS1 or SFS), and have mentioned it in the text on page 7, lines 198-199: “To do this, we first identified the highly differentiated (top 0.5% F_{st}) SNPs and indels that overlapped with a selective sweep in the NFS1 or SFS flies (Supplementary Data 4).…”

2. The meaning of F_{st} (presumably, the fixation index) should be clearly defined.

The fixation index measures the amount of genetic differentiation between different populations. This has been added to the manuscript on page 7, lines 180-181.

3. The highly differentiated SNPs are defined as having 0.5% top F_{st} . Is the F_{st} calculated on a single site? If so, does this criterion mean that only one allele is present in one of the groups? How do the enrichment numbers look if one considers only SNPs in which one allele is seen in all of group A flies and the other allele in all of group B flies (perfect separation)?

Yes, the F_{st} is calculated on a single site. This criterion does not mean that only one allele is present in one of the groups (which would be $F_{st} = 1$). Instead, it corresponds to an F_{st} of approximately 0.766. For reference, the SNP that we found to be correlated with *prominin* editing levels (which was present in 7/8 NFS1 lines and in none of the SFS lines) had $F_{st} = 0.803$.

If we only consider SNPs in which there is perfect separation between the groups ($F_{st} = 1$), for whole body genes, we observe 12/493 (2.4%) differentially expressed genes have a SNP, versus 146/13394 (1.1%) non-differentially expressed genes (p-value = 0.011). For head genes, we observe 13/424 (3.1%) differentially expressed genes have a SNP, versus 135/10900 (1.2%) non-differentially expressed genes (p-value = 0.0037). Please note that we changed our methods for this analysis slightly and used GSeq to calculate the p-values, which takes into account gene length (see updated “SNP enrichment in differentially expressed genes” Methods section for details, page 18).

4. What about the 9 genes in sweep regions – pls add to Supp table 1 the information on how many WT and mutant alleles were seen in each group (NFS1 and SFS). Most importantly, please specify in figure 4a which allele was observed in each of the flies at the SNP that seems to determine the editing level.

We have added the average NFS1 and SFS mutant SNP frequencies to the table. For Figure 4A, we have bolded the sample names that have the SNP mutation, and explained this in the figure legend.

5. P.7 lines 195-198: It seems the order of the sentences should be reversed: "In addition, the RNA-seq data ... and intergenic sites. We identified ..."

The manuscript has been updated to reverse the order of the sentences and is re-worded on page 8, lines 219-223: "We also calculated differentially edited sites from the RNA-seq data of fly head tissue,...".

6. Regarding sites 770 and 773 – as these are adjacent, it is likely that the mmPCR data should allow examination of the distribution of the four possible pairs of aa's (NY, NC, SY, SC). Comparing this distribution between the groups may provide more information and increased statistical significance.

We thank the reviewer for the suggestion. We carried out the analysis (Figure R6 below).

Figure R6. The proportion of the different editing isoforms for the NFS1 and SFS flies, ordered from lowest to highest editing of site 770. "NY" represents both site 770 and site 773 unedited, "NC" represents site 770 unedited and site 773 edited, "SY" represents site 770 edited and site 773 unedited, and "SC" represents both site 770 and site 773 edited.

For the NFS1 samples, the isoform with both sites 770 and 773 being unedited shares the highest proportion. The isoform with only site 773 being edited shares the next highest proportion, and the remaining isoforms are relatively much lower. For the SFS samples, the isoform with both 770 and 773 being edited shares the highest proportion. The isoform with site 770 and 773 being unedited shares the next highest proportion. The remaining isoforms (with only site 770 or site 773 being edited) are relatively much lower. Thus, it appears that the distributions of Prom editing are different between the NSF1 and SFS samples, in addition to the individual Prom

editing levels being different. We have added this as a supplementary figure (Supp. Figure 8), and added to the text on page 9, lines 265-266: “Interestingly, the distribution pattern of the different editing isoforms also differed between the NFS1 and SFS flies (Supp. Fig. 8).”

7. P.8 line 231 – please expand and provide evidence supporting the claim that most SNPs regulating editing are in cis. If this is correct, it's quite interesting, but did you check systematically for the possibility of SNPs in the ECS and found it to be less frequent? Can one exclude the possibility of errors in ECS identification?

Please see responses to Reviewer #1 for how many editing sites have SNPs in their ECSs. Yes, ECS identifications are just predictions, but are likely to be mostly accurate. For instance, a previous publication from our lab showed that there was a five-fold enrichment of editing in the predicted ECS regions compared with flanking control regions (Ramaswami, Deng, *et al*, *Nature Communications*, 2015).

8. One of the important points of the paper is the comparison of genetic and environmental regulation. It is only natural to expect some quantitative comparison of the differences in editing between the strains at an equal T, vs. the differences between 18C and 25C in the same strain. Which makes a bigger change? One can look at a global editing measure, and compare the changes, or count the number of differentially edited sites taking into account sample sizes, etc.

If we only examine the mmPCR-seq editing sites, and count significant sites as having q-value < 0.05 and >5% editing level difference between the NFS1 and SFS flies, we observe that more sites change with temperature (173 sites for NFS1 flies, 166 sites for SFS flies) than between populations (23 sites). We have updated the manuscript to include “However, we see more frequent, as well as larger changes in editing when we compare editing levels between flies raised at 18°C and 25°C, for both the NFS1 and SFS flies.” (page 11, lines 316-318).

9. Interestingly, the authors find that sites which show little change in editing between temperatures have significantly lower predicted free energies. What about sites that do change, but to the opposite direction (more editing in high T)?

We thank the reviewer for the suggestion. We carried out the analysis (Figure R7 below). They are similar to the plots in Figure 5c, but a blue line is added to represent sites which have higher editing (>5%) at higher temperature. Overall, they seem to have slightly higher free energies (i.e. less stability) than sites with higher editing at lower temperatures. However, there are only a few sites which have higher editing at 25°C than 18°C (10 for the NFS1 lines and 9 for the SFS lines), so it is difficult to make a strong claim about the trend of free energies.

Figure R7. Adapted from Figure 5c in the manuscript. For NFS1 and SFS flies, comparing predicted RNA structure free energy levels for sites which show <2% editing level difference between 18°C and 25°C, sites which show >5% editing at 18°C than 25°C, and sites which show >5% editing at 25°C than 18°C.

10. Fig 3C: in the legend – change "Significant sites" to "Differentially edited sites". Please add the background level of the genetic difference (as found far away from editing sites). Did you consider (in the x-axis) the distance to a *nearest* editing site?

Please see Figure R8 below, where the background genetic differentiation is added. The three black lines represent three sets of random unedited As found at least 5kb away from any editing site. We included 283 random unedited As in each set (the same number as the non-differentially edited sites). Please note that we have also updated the analysis slightly, so that differentially edited sites not only have $q\text{-value} < 0.05$, but also >5% editing level difference between the NFS1 and SFS flies.

Figure R8. Adapted from Figure 3c in the manuscript. Plot showing enrichment of differentiated SNPs near significantly differentially edited sites ($p\text{-value} = 0.030$, one-sided KS test), and three sets of random unedited As representing the background level of genetic differentiation.

As you can see from the figure, the regions surrounding differentially edited sites show higher genetic differentiation than the background. In regards to whether the x-axis represents the distance to the nearest editing site: if editing sites were clustered together (within 5kb of each other), then either the editing site with the greatest editing level difference between the NFS1 and SFS was included for the plot (for the differentially edited sites), or the editing site with the smallest editing level difference was chosen (for the non-differentially edited sites). This has now been added to the Methods section (“RNA editing SNP enrichment analysis” section, page 19).

Reviewers' comments:

Reviewer #1 (Remarks to the Author):

The authors addressed the concerns of all four reviewers providing many new analyses and incorporating many of the reviewers' suggestions into the revised manuscript.

Publication of this revised manuscript in Nature Communications is recommended.

Reviewer #2 (Remarks to the Author):

Major concerns:

Pp5-119-129: I am sorry, but I still think that the split (i.e., removal of the NFS2 and NFS3 clusters) is not justified and arbitrary. The authors say: *"that the two PCA clusters within NFS implies overall higher diversification of this group compared with the SFS".*

1: if you only consider NFS, then you have three clusters, not two;

2: I agree that this indicates diversification of North population. Then, if you have diversification in that NFS (maybe because the environmental pressure is more relaxed that in north canyon?) you cannot choose an extreme-clustered set of genotypes as representative of the population.

If you take PC2, NFS2 is farther away from SFS than NFS1. Indeed, it looks like that the Euclidean distance between NFS2 and SFS is larger than between NFS1 and SFS. In summary, removing NFS2 and NFS3 (i.e., 50% of the NFS data) is not justified. I might agree with the author about removing NFS3, but still, not sure.

In the response to my previous comment, the authors say that *"Since the flies in the NFS1 and NFS2 are distinct from each other, this decreases our power to identify differences in genes expression and editing".* This is not correct: increasing "n" always gives you more power to detect differences, if such differences exist. If you don't find differences by increasing "n" (i.e, your result is not significant any more), is because you were missing the actual standing variation of the NFS population. This is in my opinion a critical aspect. The MS must focus on the 16 NFS lines, and maybe include in the supplement the SFS v NFS1 analyses.

This point also applies for the detection of the selective sweeps and all other analyses. If the authors use a non-random sample (i.e., NFS1) that do not represent the standing genetic variation of the whole NSF population, the selective footprints might be artifactual. The authors do not show (as they do with gene expression and editing) how many sweep regions are significant when considering the whole NFS1.

Pp6 and 7: regarding the figure 2e, I still do not see the clustering. If the authors think there is clustering they should highlight then clusters in the figure 2e as they do with figure 1b. That would really help.

Regarding the appropriateness of PCA to compare genetic and gene expression differentiation between different drosophila population: the authors refer to the article by Marin et al. After reading myself the MS, I learnt that the (visual) comparison of the SNPs PCA and expression PCA is not valid per se. Marin et al. say (pp3, 2nd paragraph in the original article): *"A formal test of this hypothesis is presented in the last subsection".* In other words, Marin et al. developed a statistical test based on the coefficient of variation, randomization, etc, to properly test whether the SNP dataset correlates with gene expression. To my knowledge, the authors just compare visually the SNP and expression PCA assumes correlation, but they do not analyze the data as in Marin et al. This also applies to figure 3b: there is no proper testing of the hypothesis that editing level divergence correlates with genetic divergence.

Figure 3c: I don't understand why the authors do not include now the confidence interval as they did in the reply to reviewers. I think it is good for the readers to have this information to properly interpret the strength of the signal.

Pp18-Methods. About RNA editing comparison using RNA-seq data: I don't understand why the author do not include LRT explanation in the paper. Without this explanation, the reader cannot reproduce the paper's results.

GxT analysis: the authors say that they implemented a GLM. What model? Poisson, binomial, negative binomial, gamma...? What should the reader do to reproduce the data?

Minor concerns:

Pp5, 130: instead of "which regions of the genome" I'd suggest "which fraction of the standing genomic variations is under selection" or similar.

Pp6, 152: I think the authors want to say Supplementary Data 2 instead of Sup. Data 3.

Reviewer #3 (Remarks to the Author):

The authors did a thorough job of responding to my comments and I am satisfied with the current manuscript. The additional analysis performed on the wild caught vs lab strains was very interesting. I congratulate the authors for a very interesting and thorough study

Reviewer #4 (Remarks to the Author):

The authors have answered satisfactorily most of my questions, and I think the paper is nearly ready for publication. I have a few more minor comments/questions:

1. I still don't see enough evidence to support the claim that most SNPs regulating editing are in cis. Three (out of 26) sites have a diff. SNP in cis, compared to three that have one in

the ECS (p. 9, new version).

2. Talking about these stats - which 26 sites are being looked at here? There is one set of 60 (51+9) sites differentially edited sites; then 32 sites with >5% difference; then 23 sites after clusters are represented by one site.

3. The wording on the added caveat on p.8 lines 223-6 seems to suggest this possible artefact is specific to RNA-seq and not to mmPCR. If this implication is intentional, please explain. Otherwise, please rephrase to avoid confusion.

We thank the reviewers for the prompt processing and constructive feedback of our manuscript. Please see below for our point-by-point responses to reviewers' comments and suggestions. We paid particular attention to providing additional quantitative evidence demonstrating the separation of the NFS sub-populations, and providing new analyses comparing the NFS2 and SFS flies, based on your suggestions. The changes we have made to our manuscript are highlighted in red for your convenience.

Reviewers' comments:

Reviewer #1 (Remarks to the Author):

“The authors addressed the concerns of all four reviewers providing many new analyses and incorporating many of the reviewers' suggestions into the revised manuscript.

Publication of this revised manuscript in Nature Communications is recommended.”

Reviewer #2 (Remarks to the Author):

“Major concerns:

Pp5-119-129: I am sorry, but I still think that the split (i.e., removal of the NFS2 and NFS3 clusters) is not justified and arbitrary. The authors say: “*that the two PCA clusters within NFS implies overall higher diversification of this group compared with the SFS*”.

1: if you only consider NFS, then you have three clusters, not two;

2: I agree that this indicates diversification of North population. Then, if you have diversification in that NFS (maybe because the environmental pressure is more relaxed that in north canyon?) you cannot choose an extreme-clustered set of genotypes as representative of the population.

If you take PC2, NFS2 is farther away from SFS than NFS1. Indeed, it looks like that the Euclidean distance between NFS2 and SFS is larger than between NFS1 and SFS. In summary, removing NFS2 and NFS3 (i.e., 50% of the NFS data) is not justified. I might agree with the author about removing NFS3, but still, not sure.

In the response to my previous comment, the authors say that “*Since the flies in the NFS1 and NFS2 are distinct from each other, this decreases our power to identify differences in genes expression and editing*”. This is not correct: increasing “n” always gives you more power to detect differences, if such differences exist. If you don't find differences by increasing “n” (i.e, your result is not significant any more), is because you were missing the actual standing variation of the NFS population. This is in my opinion a critical aspect. The MS must focus on the 16 NFS lines, and maybe include in the supplement the SFS v NFS1 analyses.”

We agree with the reviewer that comparisons using the NFS2 population may still be desired, and could provide a more complete picture of how flies adapt to the environment on the NFS. Therefore, for the analyses we performed comparing the NFS1 and SFS flies, we have now performed equivalent analyses comparing the NFS2 and SFS flies. The new results are now discussed in the main text, with figures pertaining to the NFS2 and SFS comparisons in the supplementary information. Although we observe that genetic *cis*-regulation is largely guiding gene expression and RNA editing differences between the NFS2 and SFS flies, similar to what we observed between the NFS1 and SFS populations, the types of genes that are differentially expressed and sites that are differentially edited are generally quite different between the two sets of comparisons. This further suggests that the NFS1 and NFS2 populations have gone through very distinct evolutionary trajectories, despite being from a similar geographic location, and re-affirms our original claim that they should be examined as separate clusters.

Therefore, we provide additional evidence (**Figure R1**) demonstrating how the NFS sub-populations are different from one another. The number of significantly differentially expressed genes (adjusted p-value < 0.05), differentially edited sites (adjusted p-value < 0.05 and $\geq 5\%$ editing level difference), and number of SNPs ($F_{st} = 1$) are plotted for each pairwise comparison between the different population clusters in the SNP PCA (NFS1, NFS2, NFS3, and SFS). As you can see, the NFS1 and NFS2 populations share at least as many differences between each other as there are between the NFS1 and SFS, and the NFS2 and SFS.

Figure R1: For each pairwise comparison between the NFS1, NFS2, NFS3, and SFS sub-populations, plots of the number of significantly differentially expressed genes in whole bodies (a), the number of significantly differentially expressed genes in heads (b), the number of SNPs with $F_{st} = 1$ (c), and number of differentially edited sites from the mmPCR-seq data (d).

It is evident from the plots that the NFS3 population tends to share more differences between the NFS1 and NFS2 populations than when compared to the SFS. For this reason, along with our original claim based on the SNP PCA that the NFS3 likely represents recent migrants from the SFS (also supported by a previous study cited in the manuscript that observed higher migration from the SFS to the NFS than vice versa), we are not including the NFS3 flies in any of the revised analyses.

“This point also applies for the detection of the selective sweeps and all other analyses. If the authors use a non-random sample (i.e., NFS1) that do not represent the standing genetic variation of the whole NSF population, the selective footprints might be artifactual. The authors do not show (as they do with gene expression and editing) how many sweep regions are significant when considering the whole NFS1.”

We have now included information on the selective sweeps found in the NFS2 populations (Supplementary Figure 2, Supplementary Data 1).

“Pp6 and 7: regarding the figure 2e, I still do not see the clustering. If the authors think there is clustering they should highlight then clusters in the figure 2e as they do with figure 1b. That would really help.”

Here is the plot with the clusters highlighted/boxed (**Figure R2**):

Figure R2: PCA using whole body gene expression, with boxes around the different population clusters (based on Figure 2e in the original manuscript).

However, we think that highlighting the clusters here could be distracting, so we have left it as is in the manuscript. In addition, as described below, we see a statistically significant correlation between PC2 of this figure (whole body gene expression PCA plot) and PC1 of the SNP PCA, which is further quantitative evidence for the claim that the genetic divergence among the fly lines is similar to the gene expression divergence.

“Regarding the appropriateness of PCA to compare genetic and gene expression differentiation between different drosophila population: the authors refer to the article by Marin et al. After reading myself the MS, I learnt that the (visual) comparison of the SNPs PCA and expression PCA is not valid per se. Marin et al. say (pp3, 2nd paragraph in the original article): “*A formal test of this hypothesis is presented in the last subsection*”. In other words, Marin et al. developed a statistical test based on the coefficient of variation, randomization, etc, to properly test whether the SNP dataset correlates with gene expression. To my knowledge, the authors just compare visually the SNP and expression PCA assumes correlation, but they do not analyze the data as in Marin et al. This also applies to figure 3b: there is no proper testing of the hypothesis that editing level divergence correlates with genetic divergence.”

In response to the Reviewer’s comment that our separation between clusters was merely visual, we have tested correlations between the SNP PC1 points and corresponding gene expression PC1 points (except for whole body gene expression, where there is a significant correlation with PC2), and SNP PC1 and the editing PC1 points using Bonferroni-corrected Spearman rank correlations. We observe highly significant correlations:

SNP PC1-Editing PC1: $r=-0.780$ (p-value= $1e-6$)
SNP PC1-ExpressionHead PC1: $r=0.664$ (p-value= $5e-5$)
SNP PC1-ExpressionWholebody PC1: $r=0.026$ (p-value= 0.888)
SNP PC1-ExpressionWholebody PC2: $r=0.810$ (p-value= $6e-7$)

We are providing the results of this analysis in the manuscript to provide additional quantitative evidence that that the genetic differences among the Evolution Canyon populations are associated with the gene expression and editing differences (page 7, lines 193-199, and page 9, lines 267-268).

We also performed the Mantel test to determine the correlation between the SNP PCAs and the gene expression and editing PCAs, in same way as described in Martin et al (between the Fst matrix and Expression CV correlation matrix or Editing degree CV correlation matrix, 10,000 Monte Carlo repetitions). Here are the results:

Fst & Editing: $\rho=0.235$, p-value= 0.457
Fst & Head Expression: $\rho=0.606$, p-value= 0.162
Fst & Wholebody Expression: $\rho=0.429$, p-value= 0.341

We think part of the reason we do not get significant results ($P > 0.05$) is because of the limited number of populations/pairwise comparisons. We had 6 pairwise comparisons

compared to 21 pairwise comparisons in Martin et al. We also note that despite the higher number of comparisons in Martin et al., their result of Mantel test is not statistically significant either ($P > 0.05$).

“Figure 3c: I don’t understand why the authors do not include now the confidence interval as they did in the reply to reviewers. I think it is good for the readers to have this information to properly interpret the strength of the signal.”

We have now included the confidence interval in this plot, as well as for the equivalent plot comparing NFS2 and SFS (Supplementary Figure 13).

“Pp18-Methods. About RNA editing comparison using RNA-seq data: I don’t understand why the author do not include LRT explanation in the paper. Without this explanation, the reader cannot reproduce the paper’s results.”

We have added this to the Methods section (“RNA editing comparison using RNA-seq data”, page 21).

“GxT analysis: the authors say that they implemented a GLM. What model? Poisson, binomial, negative binomial, gamma...? What should the reader do to reproduce the data?”

In fitting the generalized linear model, we used a Gaussian distribution. This Methods section has been updated with this information (“Genotype-Environment Interaction Analysis”, page 23).

“Minor concerns:

Pp5, 130: instead of “which regions of the genome” I’d suggest “which fraction of the standing genomic variations is under selection” or similar.”

We have updated the manuscript accordingly (page 5, line 133-4: “Next, we wanted to determine which fraction of the standing genomic variation is under selection, and thus likely to be adaptive.”)

“Pp6, 152: I think the authors want to refer to Supplementary Data 2 instead of Sup. Data 3.”

We have changed this to say Supplementary Data 2 (page 6, line 157).

Reviewer #3 (Remarks to the Author):

“The authors did a thorough job of responding to my comments and I am satisfied with the current manuscript. The additional analysis performed on the wild caught vs lab strains was very interesting. I congratulate the authors for a very interesting and thorough study”.

Reviewer #4 (Remarks to the Author):

“The authors have answered satisfactorily most of my questions, and I think the paper is nearly ready for publication. I have a few more minor comments/questions:

1. I still don't see enough evidence to support the claim that most SNPs regulating editing are in cis. Three (out of 26) sites have a diff. SNP in cis, compared to three that have one in the ECS (p. 9, new version).”

If a SNP regulates the editing levels of a site in cis, we consider that SNP to be part of the same transcript as the editing site. So, if a SNP is in the ECS of a differentially edited site, we consider that to be regulating editing “in cis”; therefore, both types of analyses (examining the number of sites with SNPs nearby, and SNPs in their ECSs) would provide information on genetic *cis*-regulation.

“2. Talking about these stats - which 26 sites are being looked at here? There is one set of 60 (51+9) sites differentially edited sites; then 32 sites with >5% difference; then 23 sites after clusters are represented by one site.”

Both the 26 sites and the 23 sites come from the 32 sites that are significantly differentially edited and have at least 5% editing level difference. The reason the number of sites are slightly different after taking into account clustered sites is because different cluster sizes are used, depending on the analyses. For the 23 sites (represented in the plot in Figure 3c), the cluster size used was 5kb, since that is distance represented on either side of the editing site in the plot. For the 26 sites, the cluster size used was only 100 bases, since we only examined SNPs within 100 bases of or in the ECSs of this set of sites.

“3. The wording on the added caveat on p.8 lines 223-6 seems to suggest this possible artefact is specific to RNA-seq and not to mmPCR. If this implication is intentional, please explain. Otherwise, please rephrase to avoid confusion.”

The caveat also applies to mmPCR-seq editing level differences. The manuscript has been updated accordingly (page 9, line 251).

REVIEWERS' COMMENTS:

Reviewer #4 (Remarks to the Author):

The authors have adequately answered the questions raised by the referees.

The only point (I think) was missed is the question regarding Martin et al. In my opinion, the methods introduced in that paper (Fig.5 and paragraph starting 'We next used established methods to assess the proportion of gene expression variation among individuals attributable to population identity') are the ones which are relevant to the present study.

However, since the authors of the present study were not interested in assessing significance per gene, I don't really follow the logic of the question to begin with, and therefore the answer given is satisfactory. That is, the significant correlations between the SNP PC's and the expression PC's are convincing.